# Enhancer histone-QTLs are enriched on autoimmune risk haplotypes and influence gene expression within chromatin networks

Richard C. Pelikan [1], Jennifer A. Kelly[1], Yao Fu[1], Caleb A. Lareau [2,3], Kandice L. Tessneer[1], Graham B. Wiley[1], Mandi M. Wiley[1], Stuart B. Glenn[1], John B. Harley[4], Joel M. Guthridge[5], Judith A. James[5,6], Martin J. Aryee [2,3], Courtney Montgomery[1] & Patrick M. Gaffney[1]

Genetic variants can confer risk to complex genetic diseases by modulating gene expression through changes to the epigenome. To assess the degree to which genetic variants influence epigenome activity, we integrate epigenetic and genotypic data from lupus patient lymphoblastoid cell lines to identify variants that induce allelic imbalance in the magnitude of histone post-translational modifications, referred to herein as histone quantitative trait loci (hQTLs). We demonstrate that enhancer hQTLs are enriched on autoimmune disease risk haplotypes and disproportionately influence gene expression variability compared with non-hQTL variants in strong linkage disequilibrium. We show that the epigenome regulates HLA class II genes differently in individuals who carry HLA-DR3 or HLA-DR15 haplotypes, resulting in differential 3D chromatin conformation and gene expression. Finally, we identify significant expression QTL (eQTL) x hQTL interactions that reveal substructure within eQTL gene expression, suggesting potential implications for functional genomic studies that leverage eQTL data for subject selection and stratification.

[1] Division of Genomics and Data Sciences, Arthritis and Clinical Immunology Research Program, Oklahoma Medical Research Foundation, Oklahoma City 73104 OK, USA. [2] Department of Biostatistics, Harvard T.H. Chan School of Public Health, Charlestown 02115 MA, USA. [3] Department of Pathology, Harvard Medical School, Boston 02115 MA, USA. [4] Center for Autoimmune Genomics and Etiology, Cincinnati Children's Hospital, Cincinnati 45229 OH, USA. [5] Arthritis and Clinical Immunology Research Program, Oklahoma Medical Research Foundation, Oklahoma City 73104 OK, USA. [6] Departments of Medicine and Pathology, University of Oklahoma Health Sciences Center, Oklahoma City 73104 OK, USA. Correspondence and requests for materials should be addressed to P.M.G. (email: Patrick-Gaffney@omrf.org)

A fundamental objective of human genetics is to understand how genotypes influence phenotypes. To this end, genome-wide association studies (GWAS) have successfully identified thousands of convincing and reproducible statistical associations between genetic variants, phenotypic traits, and diseases in humans[1]. GWAS data, however, do not carry fundamental information about how the flow of genomic information from genotype to phenotype is acted upon by the epigenome, potentially limiting the effectiveness of translating GWAS data into actionable clinical knowledge for diagnosis, prognosis, and prediction of complex diseases in patients. Thus, in the post-GWAS era, significant effort has been directed toward characterizing epigenetic states, and the mechanisms by which the epigenome orchestrates the flow of genomic information in specific cellular contexts[2-4]. These studies have demonstrated that much of the non-protein-coding genome is dedicated to epigenomic activity[2], and that specific post-translational modifications (PTMs) on histones can define the location and functional state of enhancer elements and regions of the genome that are transcriptionally activated or inhibited[3]. Moreover, the epigenome coordinates information flow in three-dimensional (3D) space through chromatin loops that facilitate long-range engagement of enhancers with promoters of genes whose expression sustains or modulates the cell state[4].

From this framework, it follows that DNA mutations and polymorphisms have the potential to modify cellular phenotypes by inducing changes in the epigenome circuitry and how it processes information, particularly for complex genetic diseases. Accordingly, the majority of GWAS variants locate to regions of non-protein-coding DNA[5,6] and are enriched in enhancer elements that function as epigenome modulators of gene expression[4,6]. Genetic variants can induce epigenetic "footprints"—manifested as allele-specific imbalances in the magnitude of histone PTMs (histone quantitative trait loci (hQTLs))—that identify functional states of enhancer elements[7]. These hQTLs can disrupt transcription factor binding motifs leading to enhancer dysfunction that is heritable from parent to offspring[8,9]. These results suggest that a priori knowledge of epigenome alterations induced by hQTLs could focus analysis of disease risk haplotypes on enhancer elements most likely to harbor disease-modifying variants, even within the context of strong linkage disequilibrium (LD). In addition, knowledge of hQTLs and their effects on quantitative gene expression traits (eQTLs), particularly in the context of the 3D chromatin network, could improve the precision of this widely used method of genotype-to-phenotype analysis.

To quantify the impact of hQTLs on complex disease risk haplotypes and gene expression traits, we performed a genome-wide screen in 25 lymphoblastoid cell lines (LCLs) from European-American patients with systemic lupus erythematosus (SLE) to identify hQTLs in weak and strong enhancers defined by the presence of H3K4me1 or H3K27ac, respectively[10,11]. Our results show that enhancer hQTLs are significantly enriched in autoimmune disease risk haplotypes and exert a disproportionate influence on gene expression variability when compared with non-hQTL variants in strong LD with them. We show that the HLA class II locus is densely populated with enhancer hQTLs, resulting in differential 3D chromatin conformation and gene expression between the two most common HLA class II autoimmune disease risk haplotypes—HLA-DR3 and HLA-DR15. Finally, we identify statistically significant physical interactions between eQTLs and hQTLs, in LD, that modify eQTL-based gene expression and explain, in part, gene expression variability of eQTL data, suggesting potential implications for functional genomic studies that leverage eQTL data for subject selection and stratification.

## Results

**Genome-wide scan identifies 6261 enhancer hQTLs.** Chromatin immunoprecipitation (ChIP) sequencing peaks that were reproducibly measured across two technical replicates in at least 13 of 25 LCLs were used to construct a consensus peak map specific for each chromatin mark. In total, we detected 33,437 H3K27ac and 39,613 H3K4me1 consensus peaks, of which 27,404 overlapped both marks.

Genotyping each cell line using the Illumina HumanOmni 2.5 M SNP array resulted in 1,468,562 variants that passed quality control. Imputation with the 1000 Genomes Phase 3 reference panel[12] produced a total dataset of 9,603,466 SNPs. Of these, 315,210 heterozygous SNPs were located within a H3K27ac or H3K4me1 consensus enhancer peak. Following realignment of ChIP-seq reads using WASP to control for reference genome alignment bias[7], the 315,210 SNPs were tested for allelic imbalance on histone PTMs using the combined haplotype test (CHT)[8]. Comparison of the observed read count distribution with permuted read counts demonstrated that the CHT was well calibrated for both histone marks (Supplementary Figure 1). In total, we identified 6261 significant hQTLs (2007 with $r^2 \leq 0.8$) distributed throughout the genome; 5829 significant at the 10% family-wise error rate (FWER)-adjusted $p$-value (nominal $p$-value $< 8.9E-07$), and 432 suggestive at 20% FWER (nominal $p$-value $< 9.8E-07$) (Fig. 1a, b, Supplementary Data 1). H3K27ac hQTLs were strongest in the HLA region, with greatest significance located between *HLA-DRB1* and *HLA-DQA1* ($p = 1.23E-142$) (Fig. 1a). While strong H3K4me1 hQTLs were also observed in the HLA region (HLA-DPB2; $p = 3.13E-24$), the most significant H3K4me1 hQTLs were located at *LRRC16A* ($p = 7.82E-40$), ~4 Mb upstream of the major histocompatibility complex (MHC) locus (Fig. 1b). The most significant non-HLA hQTLs ($p < 1E-45$) for H3K27ac were observed with *CACNA1E* (chr 1), a region between *SLMAP* and *FLNB* (chr 3); *TEC* (chr 4); *ANTXRLP1* (chr 10); *OAS1* (chr 12); and *TSHZ1* (chr 18) (Fig. 1a). Strong H3K4me1 non-HLA hQTLs ($p < 1E-24$) were observed in a region between *ZBTB18* and *C1orf100* (chr 1); *RNASEH1* (chr 1); *PNOC* (chr 8); *TIMM23B* (chr 10); *CHST11* (chr 12); and *PLCG2* (chr 16) (Fig. 1b).

To replicate our findings, we utilized publicly available genotyping data and LCL H3K27ac ChIP-seq data[9,13] to identify hQTLs in ten independent Caucasian subjects. Of the 6261 H3K27ac hQTLs in our discovery set, 5068 hQTLs in the replication set had sufficient read depth to test for allelic imbalance. Following our CHT analysis, we observed 2181 (43%) hQTLs in the replication set with evidence of allelic imbalance at $p < 0.05$ (Supplementary Data 1; Supplementary Table 1). Given that the replication dataset was only 40% the sample size and had less than one third the sequencing read depth of our discovery sample (Supplementary Figure 2), we interpret these results as strong support for the reproducibility of our LCL hQTLs.

We defined the effect size (ES) for hQTLs as the alpha parameter/beta parameter ratio produced by the CHT, where alpha was the maximum likelihood estimate of the reference allele read count and beta was the maximum likelihood estimate of the alternative allele read count. We observed an average $\log_2(ES)$ of 0.89 for all hQTLs, with stronger effects observed with non-HLA variants ($\log_2(ES_{average}) = 0.90$; $\log_2(ES_{max}) = 3.57$) compared to HLA variants ($\log_2(ES_{average}) = 0.83$; $\log_2(ES_{max}) = 2.47$) (Fig. 1c; Supplementary Data 1). These results demonstrate that our hQTLs produced strong effects on allelic imbalance with variants having, on average, almost two-fold more histone reads with one allele versus the other.

The distribution of hQTLs was skewed such that 4858 (78%) were unique to the H3K27ac mark and located in 879 of the

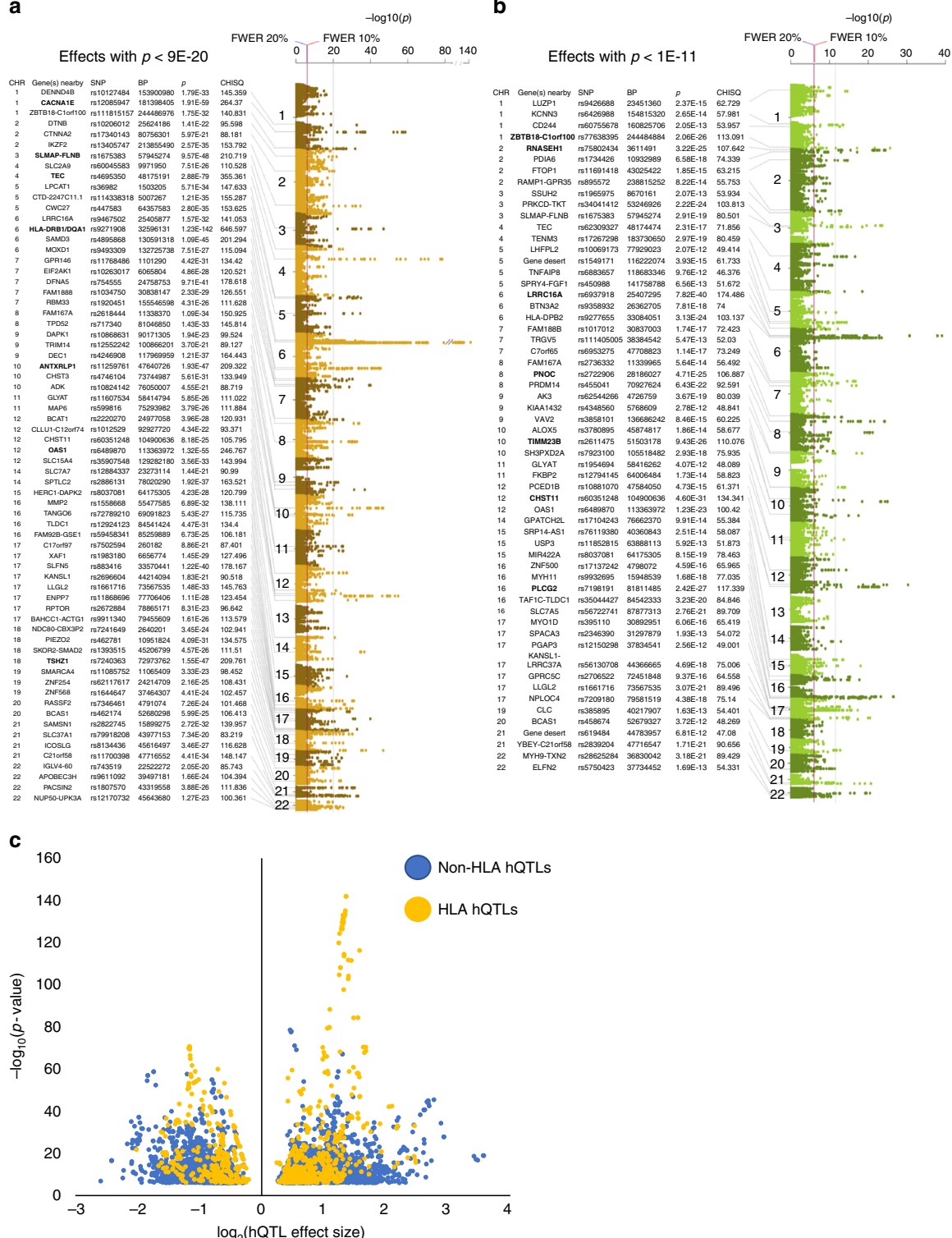

**Fig. 1** hQTL Manhattan plots and plot of hQTL effect sizes. Top effects for **a** the H3K27ac mark ($p < 1E{-}30$, vertical gray line), and **b** the H3K4me1 mark ($p$ , $< 1E{-}15$, vertical gray line) are indicated. Strongest effects for each mark are in bold. Red vertical line indicates the 10% FWER threshold (nominal $p$ , $=$ , 8.9E$-$07); blue vertical line indicates the 20% FWER threshold (nominal $p = 9.8E{-}07$). Y-axis is the $-\log_{10}(p)$ from the CHT; chromosome position is plotted on the x-axis. **c** Volcano plot of the hQTL effect sizes. The $\log_{2}$(effect size) is plotted on the x-axis where values >0 had more reference than alternate allele specific reads and those with effect size values <0 had more alternate than reference allele specific reads. Y-axis is the $-\log_{10}(p)$ from the CHT. hQTLs located in the HLA region are in yellow and non-HLA hQTLs are in blue

H3K27ac consensus peaks, while 817 (13%) were unique to H3K4me1 and found in 628 of the H3K4me1 consensus peaks. Only 586 (9%) hQTLs were in consensus peaks shared by both marks. Consensus peaks containing hQTLs were enriched, when compared to the non-hQTL consensus peaks, for chromatin states consistent with enhancers, as well as transcription factor motifs found in the B lymphoid lineage (Supplementary Figure 4; Supplementary Data 2).

**hQTLs map to disease risk haplotypes and impact gene expression**. To add disease specific context to our hQTL dataset, we utilized 1436 index SNPs from 21 autoimmune (AI) diseases (Supplementary Table 2) reported in the NHGRI-EBI Catalog of Published Genome-Wide Association Studies (October 17, 2016; www.ebi.ac.uk/gwas) to construct risk haplotypes based on LD ($D$, $′ \geq 0.8$). We found that our hQTLs were enriched on risk haplotypes, likely identifying enhancers at risk for regulatory dysfunction. A total of 386 hQTLs mapped to 44 reported AI disease risk haplotypes ($p_{permutation} = 4.9E{-}62$; $N = 180$ hQTLs expected by chance). Interestingly, 15 hQTL SNPs were the exact index AI SNP reported in the catalog, while 344 hQTL SNPs were strong proxies ($r^2 \geq 0.8$) of an AI index SNP. Considering only SLE, we observed 68 hQTLs mapping to SLE risk haplotypes in *BLK*, *FAM167A*, *HCG27*, *HLA-DQA1*, *HLA-DRB1*, *PXK*, and *SLC15A4* ($p_{permutation} = 1.9E{-}4$; $N = 45$ hQTLs expected by chance). Altogether, 1520 hQTLs (24%) mapped to 550 risk loci from 239 different genetic traits reported in the GWAS catalog ($p_{permutation} = 2.1E{-}90$; $N = 1007$ hQTLs expected by chance). To ensure our results were not driven solely by HLA loci, we removed all HLA hQTLs from the analysis and still observed significant enrichment with AI disease ($p_{permutation} = 4.4E{-}3$; $N$, $= 177$ observed/146 expected by chance), and complex traits reported in the NHGRI database ($p_{permutation} = 4.2E{-}40$; $N = 1151$ observed/839 expected by chance), and suggestive enrichment with non-HLA SLE risk haplotypes ($p_{permutation} = 0.073$; $N$, $= 48$ hQTLs observed/38 expected by chance).

We would expect hQTLs to produce measurable fluctuations on target gene expression variance if the allelic imbalance results in altered enhancer potency. To test this possibility, we first calculated gene-level eQTLs in the gEUVADIS European LCL dataset ($N = 358$)[14] using the same methods described there but with updated RNA-seq alignment and quantification methods[15,16] and identified 2403 (74%) of the 3259 eQTLs originally reported by gEUVADIS[14]. Using these eQTLs as functional surrogates of risk haplotypes, we identified 245 haplotypes of sufficient size ($\geq$ six variants) that contained an hQTL in strong LD ($D′ \geq 0.8$) with the eQTL but not highly correlated ($r^2 < 0.6$) in the genetic information it captured. We calculated joint effect generalized linear models for each of the 245 haplotypes to determine if incorporation of the genotype at the hQTL and each variant within the haplotype block significantly contributed to expression variation of the eQTL target gene (deviance, or $D^2$). To generate the null distribution of $D^2$ and significance at each locus, we permuted the position of the hQTL at each non-QTL variant position within the haplotype. For 87 haplotypes (36%), the hQTL demonstrated significantly increased ability to model expression variation ($P_{perm} < 0.05$) of the eQTL target gene over the eQTL alone or any other non-QTL variant on the haplotype (Fig. 2). These results suggest that variants with strong allelic imbalance at histone marks can produce measurable effects on gene expression variance, even above that of known eQTLs or other non-QTL variants in tight LD with them.

We then tested how RegulomeDB, an established functional variant annotation method[17], would rank hQTLs and non-hQTLs

using the 44 AI disease risk haplotypes that contained an hQTL in strong LD ($D′ \geq 0.8$) with the lead GWAS variant. We limited the analysis to variants with low scoring RegulomeDB scores of 1 and 2, thus focusing on variants with high confidence for functional impact (Supplementary Data 3). Among the 44 haplotypes, 63 out of 386 total hQTLs (16%) compared with 424 out of 5351 total non-hQTLs (8%) scored a 1 or 2, thus demonstrating a significant enrichment of hQTLs versus non-hQTLs ranked for high likelihood of functional impact ($p = 1.88E{-}8$). Although we cannot rule out the functional contribution of non-hQTL putative causal variants based on these analyses alone, our results suggest that knowledge of hQTLs would be valuable in deconstructing LD on risk haplotypes into component SNPs in enhancers that most likely contribute to locus-specific causality.

**HLA-DR3 and -DR15 hQTLs differentially regulate HLA genes**. The HLA region spans ~3.4 MB on Chromosome 6 and is enriched for genes that orchestrate and regulate innate and adaptive immune responses. Genetic variation in the HLA Class II region remains the most reproducibly significant finding in GWAS studies of autoimmune diseases[18–27]. Our hQTL analysis revealed significant allelic imbalance within the HLA Class I and II regions for both histone marks ($N = 1063$ hQTLs) (Supplementary Figure 4), but no significant allelic imbalance in the Class III region for either mark.

The region between *HLA-DRB1* and *HLA-DQB1*, which contains the XL9 regulatory sequence that epigenetically coordinates HLA-D gene expression[28], demonstrated the most concentrated and significant evidence of H3K27ac allelic

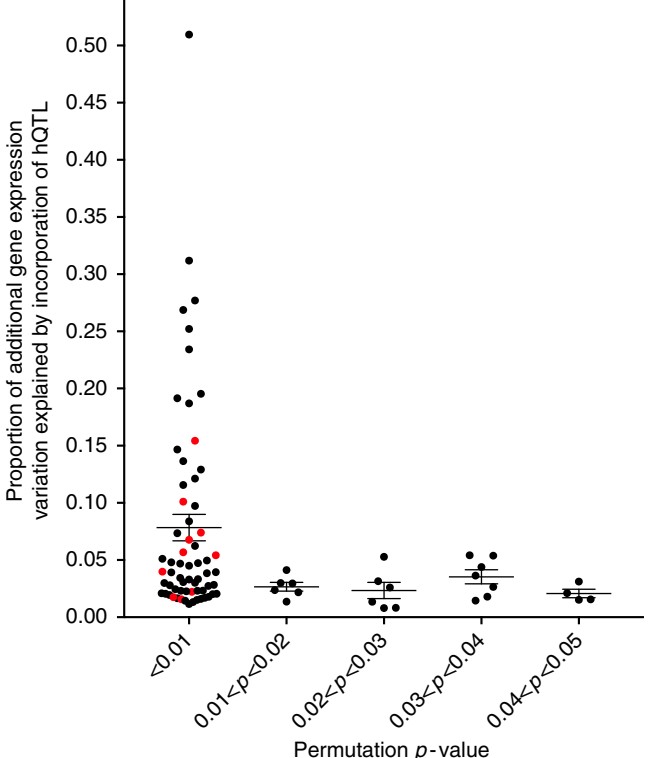

**Fig. 2** hQTLs impact gene expression in haplotype blocks that also contain eQTLs. Figure plots the 87 haplotypes for which the hQTL demonstrated significantly increased ability to model expression variation of the eQTL target gene ($p_{permutation} < 0.05$). The proportion of additional gene expression variation by incorporation of the hQTL genotypes ($D^2$) is plotted on the *y*-axis and the permutation *p*-value is plotted on the *x*-axis. HLA haplotypes are colored in red and non-HLA haplotypes are in black

imbalance at 32.596 MB ($p = 1.23E-142$) (Fig 1a; Supplementary Figure 4; Supplementary Data 1). For SLE, two HLA risk haplotypes (HLA-DR3 and HLA-DR15 (historically named HLA-DR2)) confer non-symmetrical risk for disease, with HLA-DR3 being a more potent risk factor[29–31]. In addition, eQTLs carried on these haplotypes show increased expression of HLA-DR and HLA-DQ genes compared to non-risk HLA haplotypes[28]; however, it is not known if eQTL effects differ when the two risk haplotypes are compared to each other. To determine if SLE risk haplotypes contributed differential allelic imbalance in H3K27ac enhancer marks, we imputed the HLA classical alleles in our 25 LCL samples and performed haplotype specific hQTL analyses (Fig. 3, Supplementary Figures 5-7); the majority of individuals ($N = 21$) were heterozygous for either HLA-DR3 ($N = 9$) or HLA-DR15 ($N = 12$). Since the human reference genome is homozygous for HLA-DR15[32], we wanted to ensure that any observed differences between the two haplotypes reflected the true intrinsic effect of genetic variation and were not due to alignment bias of sequencing reads. We constructed a custom genome containing the COX (HLA-DR3) Major Histocompatibility Complex (MHC) sequence in place of the PGF (HLA-DR15) MHC sequence in the human reference genome and mapped the sequencing reads from the HLA-DR3 and HLA-DR15 subjects to their respective HLA reference genomes to evaluate haplotype-specific effects. Focusing on the HLA-DRB1 to HLA-DQB1 region, we observed three strong hQTL signals (A: 32.568–32.571 Mb, B: 32.578–32.582 Mb and C: 32.588–32.606 Mb (Fig. 3; Supplementary Figure 5). Signals A and B were produced primarily from the HLA-DR15 haplotype with over three-fold more total allele-specific reads (ASR) than the HLA-DR3 haplotype (mean $ASR_{(Total)}/SNP = 49$ vs. 15% for HLA-DR15 and HLA-DR3 subjects, respectively) (Supplementary Data 4). Signal C, while exhibiting comparable total ASR between the two haplotypes, had more hQTLs with highly significant CHT p-values coming from the HLA-DR3 haplotype compared to the HLA-DR15 haplotype ($N = 261$ vs. $N = 110$) (Supplementary Data 5). To understand why the HLA-DR3 subjects had more significant hQTLs when the total ASR between the two haplotypes was comparable, we separated the total ASR into reference and alternate ASR. When mapped to the COX reference genome, the HLA-DR3 subjects had significantly higher $ASR_{(ref)}/ASR_{(alt)}$ ratios at strong hQTLs ($ASR_{(ref)}/ASR_{(alt)} = 21.35$) compared to HLA-DR15 subjects mapped to the PGF genome ($ASR_{(ref)}/ASR_{(alt)} = 2.35$) (Mann Whitney $p = 1.4E-05$). Since the alternate allele on the PGF haplotype is the reference allele on the COX haplotype for most of these variants, these results demonstrate stronger allelic imbalance with the variant alleles carried on the COX haplotype, which results in more significant hQTLs within the DR3 subjects at signal C (Supplementary Data 5).

To determine if the HLA-DR15 and HLA-DR3 haplotypes correlated with local gene expression, we measured the expression of Class II genes stratified by HLA risk haplotype in our 25 LCLs. HLA-DR15 carriers demonstrated a significant increase in expression of HLA-DQB1 (Fig. 4), whereas the HLA-DR3 carriers demonstrated significant increases in expression of HLA-DRB1 (Fig. 4). Imputing the classical HLA alleles in the European gEUVADIS dataset and conducting the same analysis provided confirmation of these results in a more powerful dataset. In addition, HLA-DR15 carriers within the gEUVADIS dataset ($N = 69$) exhibited a significant increase in expression of HLA-DQA1 compared to HLA-DR3 carriers ($N = 54$) (Fig. 4).

To determine if the genetic variation resulting in differential hQTL allelic imbalance between HLA risk haplotypes influences the chromatin topology in the HLA class II region, we used high-throughput sequencing of chromosome conformation capture coupled with chromatin immunoprecipitation (HiChIP) to develop 3D chromatin topology maps for H3K27ac and CTCF-mediated loops from independent LCLs heterozygous for HLA-DR3 or HLA-DR15 ($N = 3$ for each haplotype). Again, sequencing reads were aligned to the appropriate COX or PGF genome to minimize reference genome bias in the MHC locus. In aggregate across all six cell lines, we identified 175,833 H3K27ac and 86,891 CTCF high-confidence chromatin loops with at least four supporting paired-end tags (PETs) spanning two anchors; 8345 loops were shared by both H3K27ac and CTCF. A total of 5608 (90%) hQTLs were located within loop anchors. As expected, since our hQTLs were H3K27ac-biased, 5196 hQTLs were found only in H3K27ac-mediated loops, while 37 hQTLs were present only in CTCF-mediated loops, and 375 in both loop types.

A strong CTCF loop extended from BTNL2 to a region between HLA-DQB1 and HLA-DQA2 (Fig. 5a). This loop likely defines an insulated neighborhood[33,34] and is supported by Hi-C data from the GM12878 cell line, which demonstrates a topologically associating domain (TAD)[35,36] in this region (Supplementary Figure 8). Within this insulated neighborhood, we observed differential H3K27ac-mediated chromatin loop frequencies between the HLA-DR3 and HLA-DR15 subjects. Higher loop frequencies (as determined by PET counts) were observed within the HLA-DR15 subjects with PGF enhancers 2 ($p = 0.008$) and 5 ($p = 0.045$) (Fig. 5b, c). The PGF enhancer 2 corresponds to the exact positions of the preferential HLA-DR15 hQTL signals A and B described above, and the PGF enhancer 5 lies within the region of HLA-DQB1 (32.625–32.641 Mb on the human reference genome) that exhibits stronger hQTLs within the HLA-DR15 subjects compared to HLA-DR3 subjects (Fig. 3b); additionally, these two enhancers interact through a loop that is absent in the HLA-DR3 subjects (Fig. 5b, c (red loop)). Thus, the strong allelic imbalance observed in this region with the HLA-DR15 haplotype likely results in the significantly increased loop frequency to these enhancers and the increased expression of HLA-DQA1 and HLA-DQB1 within the HLA-DR15 subjects.

By comparison, the HLA-DR3 subjects demonstrated significantly higher loop frequency with the HLA-DRB1 promoter COX enhancer 1 ($p = 0.013$). This enhancer exhibited strong downstream looping interactions with the COX enhancers 3 and 4; these downstream looping interactions were absent with the HLA-DRB1 promoter enhancer on the HLA-DR15 haplotype (Fig. 5b, c (blue loops)). The COX enhancers 3 and 4 correspond to the location of the strongest HLA-DR3 hQTL signal (Fig. 3b; Signal C); therefore, the strong allelic imbalance observed with HLA-DR3 in this region may be driving the significantly increased looping frequency to the HLA-DRB1 promoter and increased expression of the HLA-DRB1 gene in the HLA-DR3 subjects. In addition, the HLA-DRB1 promoter has two additional Alu repeats present only on the COX genome in this region; otherwise, the two sequences were very similar. Some Alu repeats have the potential to modulate gene transcription[37], modulate nucleosome positioning[38] and shape the epigenomic landscape, thus these HLA-DR3 specific Alu repeats may also contribute to the preferential looping and increased HLA-DRB1 gene expression observed in HLA-DR3 subjects.

**hQTLs reveal substructure in eQTL regulatory networks.** hQTLs located within the chromatin looping network of an eQTL could potentially modify the eQTL effect if interacting on the same target gene promoter. To assess this in our dataset, we used a generalized linear model to measure the joint action of independent ($r^2 < 0.6$; $D' < 0.6$) hQTL x eQTL SNP combinations on

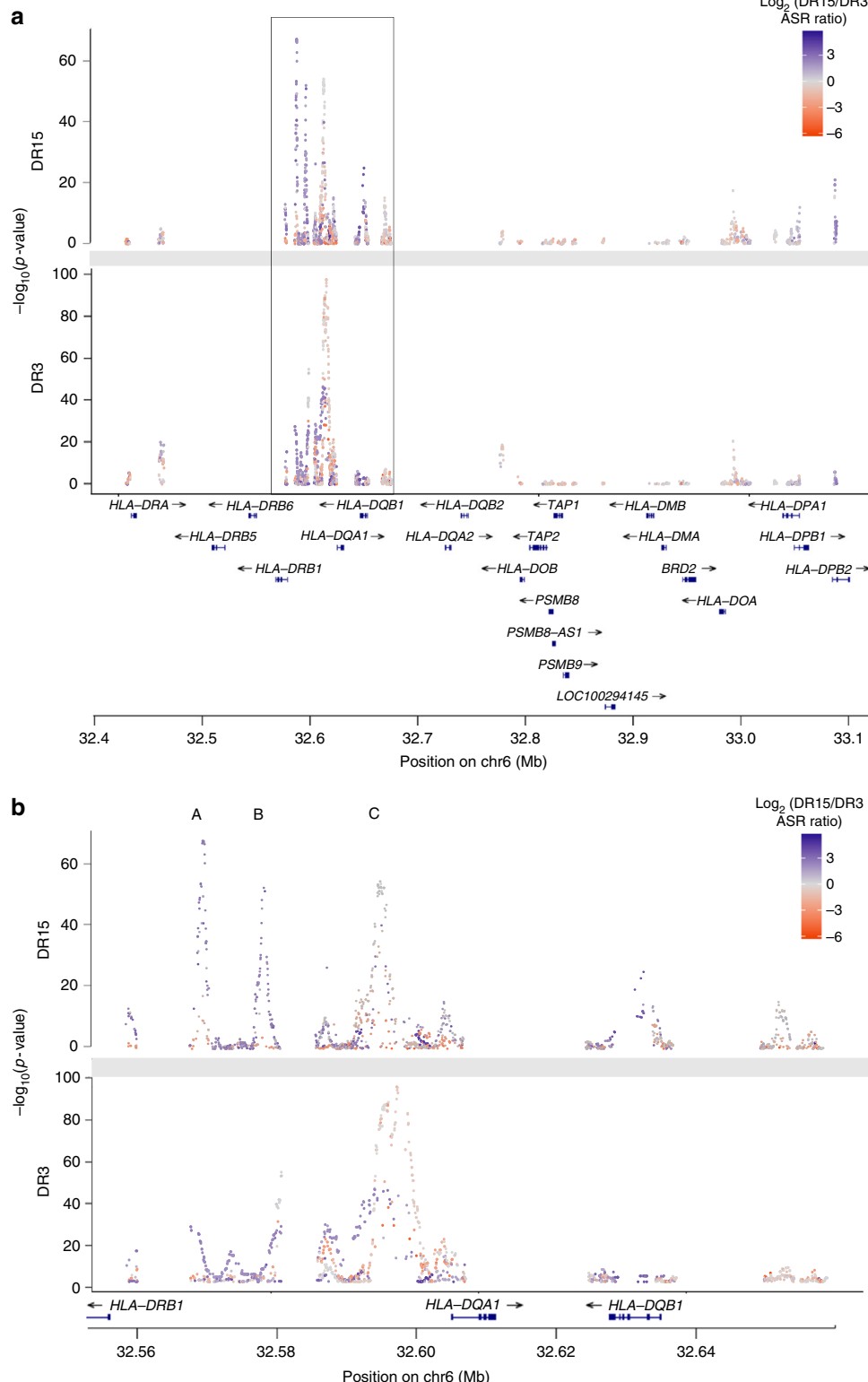

**Fig. 3** HLA Class II hQTLs have disproportionate DR15/DR3 ASE ratios. **a** Representation of HLA Class II (32.4–33.1 Mb) hQTLs by individuals heterozygous for the HLA-DR15 (top panel) or HLA-DR3 (bottom panel) haplotypes. Y-axis is the $-\log_{10}(p)$ from the CHT; chromosome position (Mb) is plotted on the x-axis. hQTLs are colored by the $\log_2$(DR15 allele specific reads/DR3 allele specific reads); dark purple values (max = 5.63) indicate more allele specific reads at a particular SNP in the HLA-DR15 and dark orange values (min = −6.11) indicate more allele specific reads in the HLA-DR3 samples at a particular SNP. **b** Zoomed in region highlighted by black box in **a** (32.56–32.66 Mb) is displayed in **b**. Signals A (32.568–32.571 Mb), B (32.578–32.582 Mb), and C (32.588–32.606 Mb) are indicated

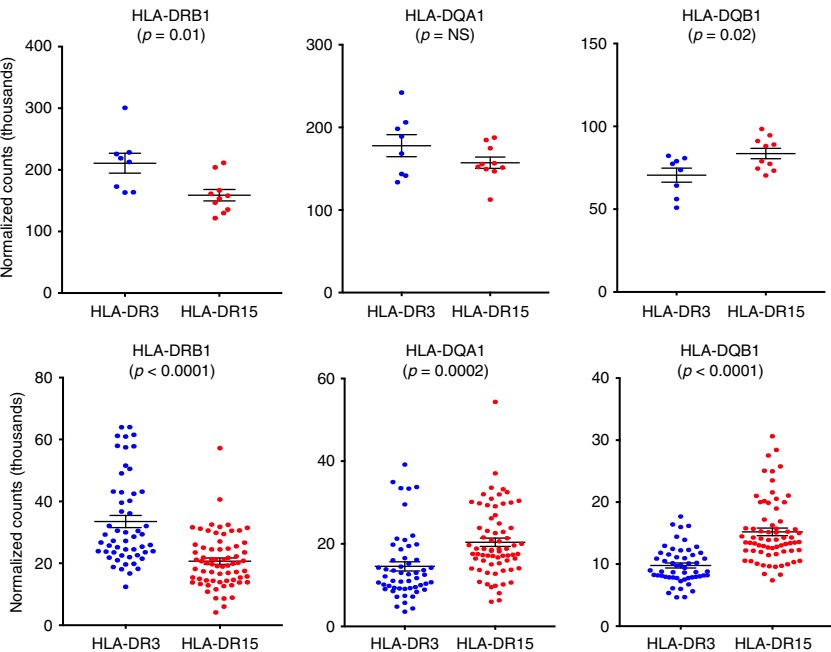

**Fig. 4** Impact of HLA-DR3 and DR15 haplotypes on HLA Class II gene expression. Data presented are from our 25 LCL samples (top panel) and the European gEUVADIS dataset (bottom panel). Normalized gene counts (measured in thousands) for each gene are plotted on the y-axis and colored by risk haplotype (DR3: blue; DR15: red); risk haplotype is presented on the x-axis. Mean and standard error of the mean bars are given in black

eQTL target genes. Following multiple hypothesis correction, we identified 17,228 significant (FDR ≤ 0.05) interactions (median/ mean distance between two SNPs: ~115 kb/~370 kb). The majority of significant interactions (N = 15,567, 90.4%) were observed with genes in the HLA region, most frequently in the Class II region (N = 13,280) and less frequently in the Class I and Class III regions (N = 1510 and 777, respectively) (Supplementary Data 6). The remaining 9.6% of significant interactions (N = 1661) involved 752 genes (Supplementary Data 7). The directionality of these modifying effects was not uniform (Fig. 6a, b). Most interactions (N = 13,506, 78%) demonstrated an opposing effect on eQTL expression in samples carrying the alternate allele at the hQTL—7279 produced repressive hQTL effects on an enhancing eQTL (Fig. 6a, b, green), while 6227 did the opposite (Fig. 6a, b, red). The remaining 3722 interactions (22%) demonstrated concordant directionality with the eQTL, i.e., expression in the same direction as the eQTL (Fig. 6a, b, blue and yellow).

In general, we found that significant gene expression substructure was revealed only when considering the joint contribution of genotypes at both the hQTL and eQTL; this substructure was not evident when evaluating the effect of the eQTL alone. We highlight *ICOSLG* and *BLK* as representative examples (Fig. 6c, d). There is a strong eQTL (rs56124762) within the Crohn's disease and ulcerative colitis-associated *ICOSLG* locus[39–41] in which individuals with increased dosage of the alternate G allele exhibited significantly higher levels of *ICOSLG* gene expression, on average, than those who only carried the reference A allele (p = 2.7E−09) (Fig. 6c). We identified an hQTL (rs8134436) located 42 kb upstream of the eQTL that was physically associated with the *ICOSLG* promoter by long-range cis-interaction. When the eQTL and hQTL genotypes are evaluated together, gene expression substructure emerged with increasing dosage of the hQTL's alternate G allele (p-interaction = 1.11E−09). Similarly, for the *BLK* locus associated with rheumatoid arthritis, SLE, and Kawasaki disease[22,42–48], a significant decrease in expression was observed

with the eQTL rs10098664 (p = 2.7E−14) and was driven by individuals who carry an increasing dose of the alternate eQTL C allele. This variant, located within the *BLK* ninth intron, was brought into proximity with the *BLK* promoter through chromatin interactions (Fig. 6d). We identified a hQTL (rs13256690) that is positioned between *BLK* and *FAM167A*, ~79KB upstream of the eQTL, and was also physically associated with the *BLK* promoter through chromatin looping. The significant interaction of the two variants with *BLK* expression (p-interaction = 8.05E−05) revealed enhanced reduction of *BLK* expression in individuals with increasing alternate allele dosage at the hQTL. Lowest *BLK* expression was observed in individuals who were homozygous for the alternate alleles at both loci. These results illustrate that an important component of variance in eQTL genotypic expression is due to independent hQTLs located within the chromatin network of an eQTL, revealing gene expression stratification that is masked by observing eQTL data alone.

## Discussion

Causal variants on disease risk haplotypes are the molecular link connecting genotype to complex disease phenotypes; however, the precision and pace of causal variant discovery is hampered by strong LD with neutral variants. Our approach demonstrates that hQTLs are enriched in autoimmune and complex disease risk haplotypes. In addition, hQTLs disproportionately contribute to gene expression variation compared to non-hQTL variants in strong LD; when located on haplotypes with known eQTLs, the hQTL demonstrated significantly increased ability to model expression variation of the eQTL target gene over the eQTL alone or any other non-QTL variants 36% of the time. This suggests that a priori knowledge of variant-induced allelic imbalance of epigenetic PTMs in enhancer elements can facilitate identification of the most influential putative causal variants on risk haplotypes even in the context of strong LD. Current approaches to causal variant discovery use bioinformatic predictions to develop

prioritized lists of putative causal variants. However, bioinformatic algorithms lack information about allele-specific effects of most genetic variants that modulate epigenome function. We found that when using RegulomeDB, an

established bioinformatic annotation method, hQTLs had twice as many high functional RegulomeDB scores compared to non-hQTLs, demonstrating that the identification of hQTLs could help prioritize and constrain the variants that would

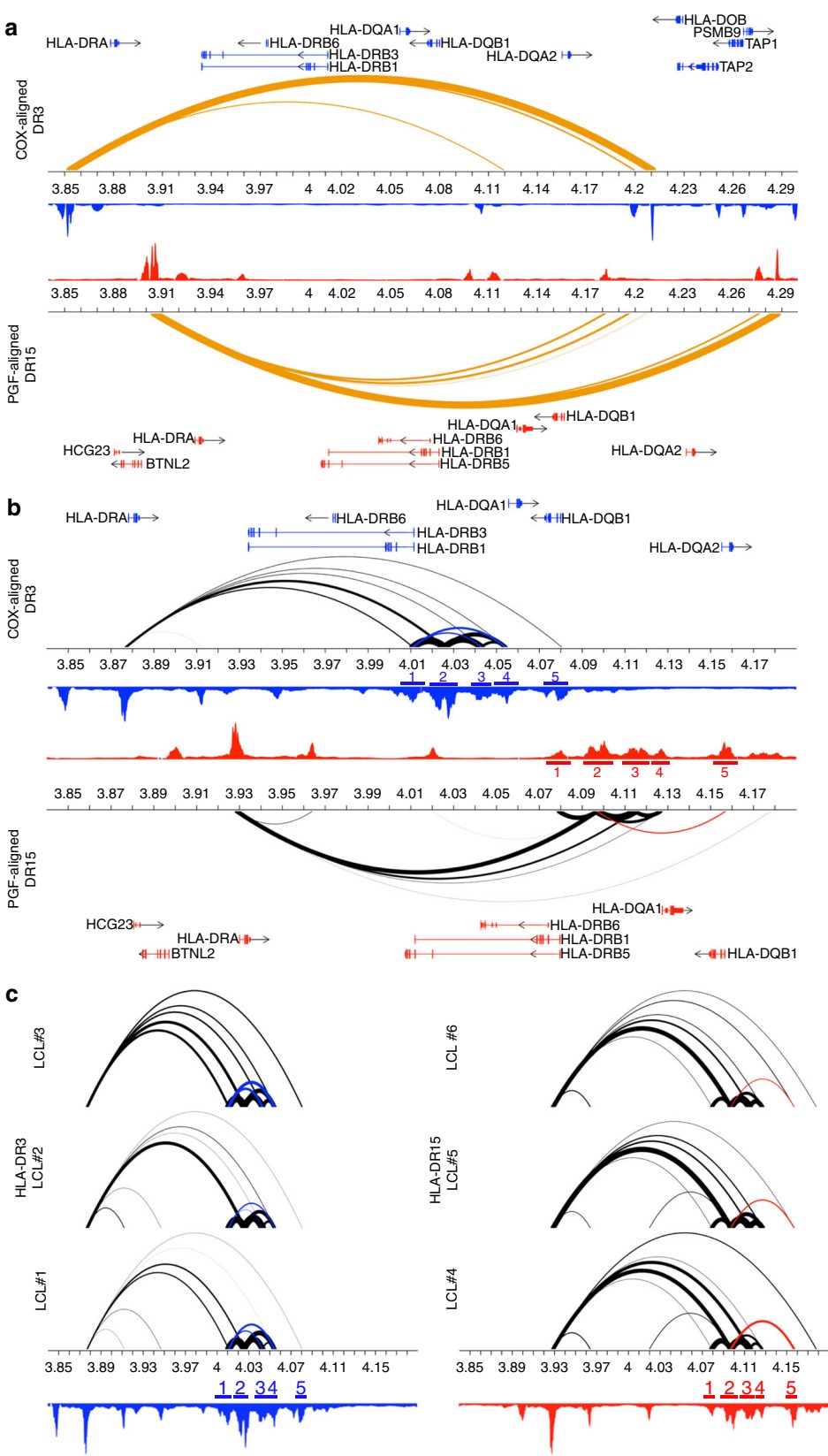

need experimental validation for mechanistic causality, thus hastening the pace for understanding how associated causal variants modulate gene expression phenotypes and risk for complex diseases.

Our results shed new light on the HLA region and how it is regulated in the context of chromatin architecture and considerable genetic variation. HLA Class II loci demonstrate allele specific gene expression when comparing HLA risk and non-risk haplotypes, and in eQTL studies without stratification on disease status[28,49]. Our results extend these observations to the level of the epigenome, where we show that the HLA Class II region exhibits significant levels of allelic imbalance in enhancer PTMs between the two most frequent HLA risk haplotypes for human SLE and other autoimmune diseases (HLA-DR3 and HLA-DR15). Using 3D chromatin topography mapping, we found that the HLA-DR15 risk haplotype demonstrated significantly increased H3K27ac-mediated chromatin loop frequencies to enhancers flanking *HLA-DQA1* and *HLA-DQB1*. These enhancers correspond to regions of strong allelic imbalance and hQTLs observed in the HLA-DR15 subjects, which are likely associated with the observed higher levels of *HLA-DQA1* and *HLA-DQB1* gene expression in these individuals. Alternatively, enhancers located in the region producing the strongest hQTLs among the HLA-DR3 individuals produced unique loops to the promoter of *HLA-DRB1* and likely contribute to subsequent higher *HLA-DRB1* gene expression compared to HLA-DR15 subjects. While the underlying mechanisms remain to be fully clarified, it is tempting to speculate that the differences in HLA class II chromatin topology between these two haplotypes and correlated gene expression of HLA-D genes may explain, in part, the increased risk for SLE conferred by the HLA-DR3 risk haplotype.

Expression QTL analysis provides a valuable intermediate phenotype for the functional characterization of causal variants on disease risk haplotypes. Moreover, as precision medicine approaches develop, eQTLs may serve as potential genetic biomarkers for selection of study subjects carrying variants that alter expression of genes encoding drug targets and pathways. However, eQTL data aggregate the effects of all forces influencing gene expression. Our data demonstrate that genetically independent hQTLs ($r^2 < 0.6$, $D' < 0.6$) within an eQTL target gene chromatin network can influence subsets of subjects for which the aggregate eQTL effect is reduced or augmented. These results confirm and extend the recent work of Corradin et al., where "outside variants" in close proximity to GWAS risk variants, but not in LD with them, contributed to variation in target gene expression and clinical risk[50]. Overall, these results highlight the complexity embedded in eQTL datasets and the need to clarify the component contributions of all variants within the chromatin network of an eQTL target locus. Defining these interactions will be important for improving the precision of studies that leverage

eQTLs for the selection of subjects for functional analysis of GWAS effects or genetically guided clinical studies.

In this report, we used an SLE-derived LCL model system to perform a genome-wide screen for allelic imbalance in PTM of enhancer histones induced by common genetic variants. Epstein-Barr Virus (EBV) transformation modifies the epigenome of B-cells resulting in promoter demethylation[51,52], more regions of open chromatin[53], and increased numbers of expressed transcripts[52,53], while maintaining most of the transcripts expressed in the original B-cell phenotype[52]. We view this as a strength of this model system since more of the genome is capable of being interrogated for allelic imbalance using our approach. Nevertheless, careful analysis of hQTLs in specific primary cell lineages and cell states will be necessary to develop a comprehensive portrait for how genetic variation modulates gene expression through allele-specific epigenome mechanisms. Moreover, hQTLs are likely to differ across ethnic backgrounds since the allele frequency distribution of SNPs varies across disparate populations. Further work will be necessary to assess the power of this approach in admixed populations.

## Methods

**Study population and cell culture.** Experiments were approved by the Institutional Review Board at the Oklahoma Medical Research Foundation (OMRF) prior to initiation. A total of 25 de-identified EBV-transformed B cell lines generated from European SLE patients (21 female, 4 male) enrolled into the Lupus Family Registry and Repository (LFRR)[54] were obtained from OMRF's Autoimmune Disease Institute Biorepository, Phenotyping, and Bioinformatics Cores and are hereafter denoted as lymphoblastoid cell lines (LCLs). All study participants provided written informed consent prior to study enrollment into the LFRR. Using previously generated genotype data, cell lines were selected on their risk haplotype repertoire for 24 SLE genes: *ATG5*, *BANK1*, *BLK*, *CD44*, *IFIH1*, *IKZF1*, *IKZF3*, *IL10*, *IRF5*, *IRF7*, *IRF8*, *ITGAM*, *JAZF1*, *LRRC18/WDFY4*, *NCF2*, *PRDM1*, *STAT4*, *TNFAIP3*, *TNFSF4*, *TNIP1*, *TYK2*, *UBE2L3*, and *XKR6*. Cell lines were maintained in RPMI 1640 medium supplemented with 10% FBS, penicillin, streptomycin, and L-glutamine.

**ChIP sequencing and analysis.** We crosslinked 10 million Epstein-Barr Virus (EBV)-transformed B cells with 1% formaldehyde for 5 min followed by careful cell lysis using the truCHIP Chromatin Shearing Kit (Covaris). Nuclei were isolated and fragmented using a Covaris E220e sonicator with the following operating conditions: Peak Incident Power: 140 Watts; duty cycle: 5%; cycles per burst: 200; treatment time: 15 min; setpoint temperature: 6 °C. Crosslinked protein/DNA was immunoprecipitated using H3K27ac (rabbit polyclonal, Abcam #ab4729, 20ug/ml) or H3K4me1 (rabbit polyclonal, Abcam #ab8895, 20 μg/ml) antibodies on Protein A + G immunomagnetic beads. Anti-acetyl histone H3 antibodies and isotype matched IgG antibodies were included as positive and negative controls, respectively. An aliquot of the fragmented chromatin, not subjected to immunoprecipitation, was also used as an input control. Chromatin crosslinks were reversed, then protein was digested with proteinase K and DNA fragments were purified using Ampure XP beads (Beckman Coulter), made into libraries, and sequenced on the Illumina NextSeq 500 (Illumina Inc., San Diego, CA) using 75 bp single-end reads. Biological replicates of each cell line were sequenced and the resulting reads were pooled. Total input DNA was also processed in replicate and pooled. To assess the quality of the ChIP-seq data, we calculated the fraction of reads in peaks (FRiP) for acCBP/p300 and RNAPII enrichment peaks and scrutinized experiments with FRiP

**Fig. 5** HLA Class II chromatin landscapes of HLA-DR3 and DR15 haplotypes. **a** CTCF HiChIP looping interactions within the HLA Class II region reveal an insulated neighborhood that extends from *BTNL2* to upstream of *HLA-DQA2*. The COX-aligned DR3 haplotype is presented on top and the PGF-aligned DR15 haplotype is presented on the bottom. Gene and bp positions are presented specific for each haplotype. Orange arcs represent CTCF-mediated HiChIP interactions. Arc thickness is proportional to the frequency of observed paired-end tags (PETs) at each enhancer. **b** H3K27ac HiChIP looping interactions within the HLA Class II region reveal differential frequency and pattern in the *HLA-DRB1* to *HLA-DQB1* region between the HLA-DR15 and HLA-DR3 haplotypes. Black arcs represent H3K27ac-mediated HiChIP interactions. Significant *p*-values are presented for enhancers that exhibit differential binding frequencies (mean PETs) for one haplotype or the other and are colored by the haplotype that exhibits higher PET counts (HLA-DR3: blue; HLA-DR15: red). H3K27ac ChIP peaks are presented in blue (HLA-DR3) and red (HLA-DR15). Haplotype specific loops are indicated in blue (COX) or red (PGF). Corresponding haplotype specific enhancers are labeled as follows—COX: 1: 4.007–4.013 Mb, 2: 4.021–4.031 Mb, 3: 4.039–4.047 Mb, 4: 4.052–4.06 Mb, 5: 4.078–4.083 Mb; PGF: 1: 4.076–4.081 Mb, *B*: 4.092–4.102 Mb, *C*: 4.111–4.120 Mb, *D*: 4.124–4.129 Mb, *E*: 4.154–4.159 Mb. **c** Individual H3K27 HiChIP looping interactions for three subjects heterozygous for HLA-DR3 (left) and three subjects homozygous for HLA-DR15 (right). H3K27ac ChIP peaks, haplotype specific loops, and haplotype specific enhancers are presented in blue (HLA-DR3) and red (HLA-DR15)

scores <1%. We also used strand cross-correlation to calculate the normalized strand coefficient (NSC) and relative strand coefficient (RSC), and repeated experiments with NSC < 1.05 and RSC < 0.8. Raw FASTQ reads were quality filtered and trimmed for Illumina adapters using Trimmomatic v0.35[55]. Reads aligning to Poly-A, Poly-C, ribosomal, mitochondrial, or Phi-X sequences were

excluded using Bowtie v2.2.4[56] with option -L 10. Filtered reads were aligned using the Burrows Wheeler Aligner (BWA)[57] to the human hg19 genome with the following options: -n 2 –o 0 –l 20. To reduce reference genome bias on read mappability, we used the WASP pipeline[7] to relocate reads potentially biased by genetic variation. Only reads overlapping single nucleotide polymorphisms (SNPs)

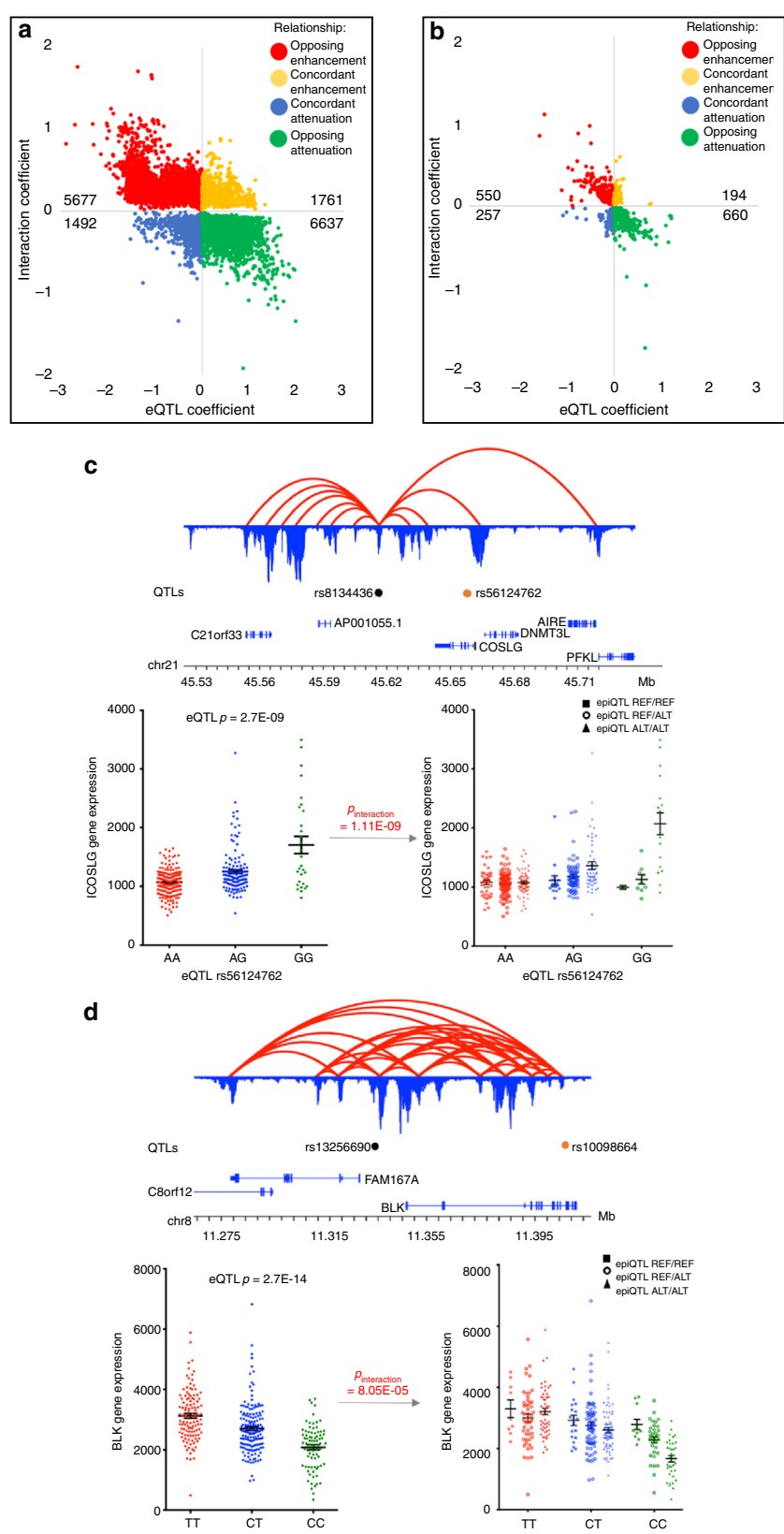

which were segregating in the genotyping data (≥2 minor alleles across population) were targeted by the realignment procedure (Supplementary Tables 3 & 4).

**Genotyping**. DNA was harvested from each LCL using standard phenol-chloroform extraction. Genotyping data for the 25 LCL samples were generated using Illumina HumanOmni 2.5–8 SNP arrays (Illumina Inc., San Diego, CA). Variants were included for analysis if they met the following criteria: genotyping rate > 0.9, minor allele frequency > 0.01, Hardy–Weinberg equilibrium ≥ 0.001. A total of 1,468,563 variants passed our filtering thresholds. Data were then pre-phased using reference information from the HapMap project phase 3 genotype map, release 3[58] together with SHAPEIT, v2.r790[59]. Imputation was carried out using IMPUTE v2.3.1[60] and the 1000 Genomes Phase 3 reference panel, v5[12]. A total of 9,603,466 SNPs resulted from the phasing and imputation process.

gEUVADIS genotyping data were obtained from the May 02, 2013 release of the 1000 Genomes project[12]. Haplotype blocks ($D' > 0.8$) were generated using PLINK2[61] from 358 Caucasian individuals using the following options:—blocks no-pheno-req –blocks-max-kb 10,000 –blocks-min-maf 0.05, using the same reference information used in the imputation of our SLE cell lines.

**RNA-seq alignment and quantification**. RNA was isolated from each LCL using TRIzol. RNA concentrations were quantified using a Qubit fluorometer, and mRNA libraries were generated using 500 ng of each RNA sample and Illumina's TruSeq Stranded mRNA Library Prep Kit. RNA-seq was performed on the Illumina HiSeq 3000 (Illumina Inc., San Diego, CA) in the OMRF's Clinical Genomics Center following Center procedures. Samples were sequenced using 75 bp paired-end reads with 12 samples per lane, which yielded ~30 million reads per sample. Primary RNA-seq FASTQ files for gEUVADIS individuals were obtained from ArrayExpress[62]. Post-sequence reads were quality filtered and trimmed for Illumina adapters using Trimmomatic v0.35[55]. Resulting reads were pseudo-aligned to coding regions of the GRCh38 genome (release 79), using Kallisto v0.43.0[15] with the following options:—bias enabled, 50 bootstraps. Expression values for 173,260 unique transcripts were measured in transcripts per million reads sequenced (TPM). To perform eQTL analyses, expression values were summarized at the gene level to transcript-length adjusted, library-size scaled counts per million (CPM) with the R package tximport[16]. Finally, to avoid false positives due to technical variation and outliers, RNA-seq expression data was normalized with PEER[63] using $K = 10$ latent factors on genes with positive expression in >50% of individuals. The PEER-derived weights were then scaled by the mean of each gene.

**HiChIP sequencing and analysis**. We crosslinked 10 million EBV-transformed B cells using 1% formaldehyde for 10 min at room temperature. Intact nuclei were isolated and digested using the MboI restriction enzyme for 4 h at 37 °C. DNA fragment ends were filled and labeled with dCTP, dGTP, dTTP and biotin-dATP. Proximity ligation was performed at room temperature overnight. Following the in situ ligation, we fragmented the ligated chromatin as previously described[64]. Crosslinked protein/DNA was immunoprecipitated using antibodies to H3K27ac (rabbit polyclonal, Abcam #ab4729, 20ug/ml) and CTCF (rabbit mAb, clone D31H2, Cell Signaling #3418, 20ug/ml) and then purified by Protein A + G immunomagnetic beads. The amount of MboI enzyme, antibody, and beads are determined by the number of cells in each sample[64]. After immunoprecipitation, DNA was eluted from the beads by incubating at 65 °C for 4 h with 5% Proteinase K. DNA was purified by Zymo DNA Clean & Concentrator Column and quantified by Qubit High Sensitivity Assay Kit. A minimum of 2 ng DNA was required for library construction. Biotin-labeled DNA fragments were further immunoprecipitated by Streptavidin M-280 Dynabeads. HiChIP libraries were generated on the streptavidin beads using the Nextera DNA Library Prep Kit. The PCR products were purified by two-sided size selection using the Ampure XP beads to capture DNA fragments between 300 and 700 bp. The integrity and quality of the libraries were assessed using an Agilent TapeStation 2200 bioanalyzer. HiChIP libraries were then pooled together and quantified using a KAPA Biosystem library quantification qPCR kit. qPCRs were performed on a Roche LightCycler 480 system using primers specific to the HiSeq flowcell. Using the template size determined from the Qubit, the pools were diluted and requantified using qPCR to double-check the starting concentrations. The pooled samples were then denatured for 5 min with NaOH. Libraries were sequenced on the Illumina NextSeq 500 sequencer on a 150-cycle paired-end Mid flowcell. HiChIP raw reads (fastq files) were aligned to the hg19

human reference genome using HiC-Pro[65]. Aligned data were processed and analyzed through the hichipper pipeline where consensus ChIP-seq tracks for either factor (H3K27ac or CTCF) were used to scaffold anchors for loop calling as recommended with default parameters[66]. Long range interactions spanning two anchor regions, termed DNA loops, were derived from linked paired-end reads that overlapped restriction fragments containing these peaks (Supplementary Table 5). Samples that passed stringent quality control had a minimum of 15% long interactions and contained intrachromosomal loops with a minimum length of 5 Kbp and a maximum length of 2 Mbp. The 3D chromatin structures generated by the HiChIP data were analyzed and visualized using the R package diffloop[67].

**Statistical analysis**. ChIP-seq peaks were called using MACS2[68] using the following options: nomodel, extsize 175. Reproducible peaks were identified on an individual basis using the Irreproducible Discovery Rate (IDR) method[69] on each pair of technical replicates. Population-wide consensus peak sets were derived from individual peak profiles by identifying genomic regions that were represented by reproducible peaks in at least 13 individuals (>50% of the population). This majority rule has been shown to perform reliably in distinguishing genuinely enriched peaks across multiple biological replicates[70]. Disjoint consensus genomic regions separated by fewer than 147 bp were merged to create the final consensus peak map for each epigenetic track.

We utilized the combined haplotype test (CHT)[8] to identify allele-dependent epigenetic footprints in ChIP-seq data. Allele-specific ChIP-seq reads were aggregated within 2000 bp regions centered around testable SNPs. SNPs evaluated by the CHT were required to have a minimum of 15 allele-specific reads in the SNP region. Only variants on autosomal chromosomes were considered for analysis by the CHT. Region-specific read counts were adjusted for GC content bias and haplotype probabilities calibrated before learning dispersion parameters for both the allele-specific and read-count portions of the CHT model. We used Holm's Family-wise Error Rate (FWER) correction procedure[71] to adjust the statistical significance given by the CHT to variants and their correlation with epigenetic read counts. We chose correcting the FWER over the more common false discovery rate (FDR) since it tends to offer better control of the Type I error rate in LD-based QTL analyses[72]. Significant hQTLs ($N = 5829$) were identified with FWER ≤ 0.1, and suggestive hQTLs ($N = 432$) with FWER ≤ 0.2. To evaluate the calibration of the CHT, we randomly permuted genotypes and their read counts at the tested variants and compared the distribution of significance values through quantile–quantile plots (Supplementary Figure 1). These distributions indicated distinct signal arising from the CHT under the true genotype and read count information, suggesting that the test was well-calibrated.

We conducted an hQTL replication study for the H3K27ac hQTLs using publicly available H3K27ac ChIP-seq data collected in LCLs from ten independent Caucasian 1000 Genomes phase three samples from two published studies[9,13]. We used the methods described above with the exception that SNPs evaluated by the CHT were required to have a minimum of 10 allele-specific reads in the SNP region due to the smaller sample size and lower ChIP sequencing depth.

Motif enrichment analysis was done using the HOMER suite v4.7[73] via the findMotifsGenome tool with the hg19 database to compare consensus peaks with and without hQTLs. Background genomic regions were specified as the list of all unique consensus peaks. We used the LOLA (Locus Overlap) tool[74] coupled with data obtained from the human Cistrome project[75] to systematically test enrichment of transcription factor (TF) binding events within hQTL-containing consensus peaks versus all consensus peaks. We used annotations from the Roadmap Epigenomics Project release 9[3] to assign GM12878 chromatin states to consensus peak regions.

To evaluate hQTLs for enrichment in catalogs of risk haplotypes, we performed SNP-based enrichment analyses as follows. First, hQTLs were binned into categories based on minor allele frequency (MAF) and distance to the nearest 5′ transcription start site (TSS). Each category represented hQTLs from a given decile of measurements of MAF and TSS distance, for a total of 100 categories. Next, for each category, we matched variant sets to hQTLs by randomly sampling an equal number of variants from those which both (a) occurred in any consensus peak region and (b) matched the criteria for MAF and TSS distance of that category. Across all categories, a total number of 337,242 variants formed the pool of matchable SNPs. Risk haplotypes were established around catalog index SNPs using the PLINK-defined LD blocks obtained from the 358 European individuals in the 1000 Genomes phase 3 dataset. The total number of matchable SNPs in risk catalogs were determined to be

**Fig. 6** hQTLs alter eQTL gene expression within chromatin networks. Quadrant plots for statistically significant interactions with genes within the HLA region (**a**) and outside the HLA region (**b**). Each statistically significant (FDR < 0.05%) distal interaction is represented by a point oriented by the value of coefficients in the generalized linear model for the eQTL main effect (x-axis) and the interaction effect (y-axis). The sign and magnitude of the coefficients determine the directionality and strength of the two effects on the eQTL target gene's expression. The total number of interactions within each quadrant are labeled. **c** Example of *ICOSLG* triplet interaction with eQTL rs56124762 and hQTL rs8134436. **d** Example of *BLK* triplet interaction with eQTL rs10098664 and hQTLs rs13256690. For both **c** and **d**, 3D topology map with H3K27ac looping data is represented by red loops. H3K27ac ChIP peaks are in blue and anchored to looping data. hQTLs (black dots), eQTLs (orange dots), and gene locations are presented. Scatterplots of the eQTL genotypes alone (left panel) and stratified by the hQTL genotypes (right panel) are given

8278 (autoimmune risk haplotypes), 2036 (SLE risk haplotypes) and 50,766 (NHGRI GWAS catalog risk haplotypes). We estimated a distribution of SNP enrichment in risk haplotypes under the null hypothesis by performing 1000 repetitions of the variant matching process and calculating the fold-enrichment of matched variants in risk haplotypes by a one-sided Fisher's exact test. We then computed fold-enrichment of the true set of hQTLs in risk haplotypes by Fisher's exact test, comparing to the null distribution to obtain a p-value.

To maximize our power for analyses using eQTL data, we obtained the raw RNA-sequencing reads from the gEUVADIS Caucasian dataset[14] and performed an eQTL analysis. Our analysis only identified 2403 (74%) of the 3259 eQTLs originally reported by gEUVADIS, which resulted from the following differences between the two analyses: (1) while we attempted to use the same analysis tools used by gEUVADIS, RNA-sequencing alignment tools have rapidly evolved and improved since the original gEUVADIS paper that utilized GEM. We, therefore, decided to use Kallisto[15] and tximport[16] for the alignment of RNA-sequencing reads. (2) The original study included 373 samples from Phase 1 of the 1000 Genomes Project in the analysis; due to sample dropout in the Phase 3 genotyping data, publicly available data included only 358 samples.

eQTL analysis within our own sample was performed with the Matrix eQTL software package (v.2.1.1) in R (v.3.2.2)[76,77]. Expression quantifications were standardized to the normal distribution to suit the assumptions of the Matrix eQTL linear model. We considered cis-eQTL interactions with less than 1 Mbp separating the SNP and the interacting transcript. Testable eQTL interactions were required to have variants with minor allele frequency ≥ 0.05 and transcripts with coefficient of variation ≥ 0.15 across gEUVADIS RNA-seq samples. We retained 214,702 eQTLs at FDR ≤ 5%. For each gene with eQTL interactions, proxy SNPs were iteratively pruned by sorting the cis-interacting SNPs by nominal p-value and calculating $r^2$ among them; SNPs with $r^2 < 0.8$ were retained.

To determine if the genotype at the local hQTL significantly contributed to expression variation of the eQTL target gene, we calculated a negative binomial generalized linear model (R function glm.nb, package MASS) with logarithmic link function to evaluate the variance explained between the lead eQTL and other non-proxy variants ($r^2 < 0.6$) within each haplotype block. If a haplotype contained multiple eQTLs or hQTLs, the lead eQTL was identified as the variant with maximum variance explained ($D^2$ statistic, generalized linear model) and the lead hQTL was defined as having the maximum nominal CHT p-value among identified hQTLs. The contribution of expression variation was estimated as:

$$D^2 = (\text{Null deviance} - \text{Residual deviance})/\text{Null deviance},$$

where the null deviance was calculated with a model without genotypic information, and residual deviance incorporates variant allele dosage (homozygote reference = 0, heterozygote = 1, homozygote alternate = 2). For the local haplotype block analysis, we ranked eQTLs by $D^2$ by modeling an eQTL's gene expression as a function of the eQTL variant's allele dosage alone. An empirical null distribution of $D^2$ was calculated for each haplotype from 10,000 permutations of hQTL positions to establish a 95% confidence interval of the p-value estimate. For both the local and distal analyses, joint effect models were fit with additive main effect terms for both eQTL and hQTL, and a multiplicative interaction term.

To rank hQTLs by functional annotation tools, we retrieved information from RegulomeDB[17] for hQTL variants and those in strong LD ($D' \geq 0.8$) with them on 44 AI disease risk haplotypes that contained an hQTL. For simplicity, we collapsed RegulomeDB scores across subcategories into a single score (i.e., 1a-1f were all scored as 1), such that RegulomeDB collapsed scores range from 8 (no RegulomeDB record) to 1 (highest functional class).

To perform haplotype-specific alignment of the sequencing data, we obtained the COX (HLA-DR3) and PGF (HLA-DR15) MHC haplotype sequences from the MHC Haplotype Project[32]. A custom genome was created by replacing the PGF sequence from the hg19 reference genome at chr6:28477798–33351543 with the COX MHC sequence. BLAT[78] was used to locate the PGF sequence endpoints. ChIP and HiChIP data for HLA-DR3 individuals were then reprocessed with the custom COX genome.

For the distal eQTL-hQTL interaction modeling, we required the eQTL and hQTL SNPs be independent of one another by LD ($r^2 < 0.6$; $D' < 0.6$), and the hQTL to be both within an H3K27ac loop anchor and the chromatin network of the eQTL target gene as defined by our 3D chromatin topography data. We modeled interaction between hQTLs and LD-independent eQTLs at an eQTL target gene as described above. Statistical significance of these models was calculated using a two-tailed z-test of the interaction coefficient. Statistical significance was controlled with the FDR. Proxies (determined by rAggr (raggr.usc.edu; $r^2 > 0.8$) using the European 1000G, Phase 3, Oct 2014; hg19 reference genome) were removed from each set of eQTLs and hQTLs prior to analysis. The sets of hQTL and eQTL variants tested for distal interactions were required to be located on separate haplotype blocks as determined by PLINK2[61]. Only interactions that had representatives for all nine eQTL x hQTL genotype combinations were considered. Distal interactions were further classified as follows: "Anchored" if both hQTL and eQTL were in the same loop anchor; "Unanchored" if the hQTL was not in a loop anchor; "Looped" if hQTL was in a loop anchor without the eQTL; "Off-target" if the loop containing the hQTL did not loop to any part of the eQTL target transcript's range (defined as 10Kbp upstream of the

transcript's transcription start site (TSS) to the transcription end site (TES)); "Joint" if the hQTL haplotype and eQTL haplotype were connected by a loop; "Disjoint" otherwise (Supplementary Figure 9).

**Data availability**. The ChIP-seq, HiChIP-seq, and RNA-seq data that support the findings of this study have been deposited into NCBI's Gene Expression Omnibus[79] and are accessible through GEO Series accession number GSE116193.

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

## Acknowledgements

We want to thank the SLE patients whose sample contributions were essential for these studies. We also thank Bryce van de Geijn and Graham McVicker for their time and excellent support with the WASP and CHT programs. Research reported in this publication was supported by the National Institute of Allergy and Infectious Diseases: U19AI082714; National Institute of Arthritis and Musculoskeletal and Skin Diseases: P30AR053483, R01AR056360, R01AR063124; and the National Institute of General Medical Sciences: P30GM110766 and U54GM104938. The content of this publication is solely the responsibility of the authors and does not necessarily represent the official views of the National Institutes of Health.

## Author contributions

P.M.G. supervised the project. R.C.P., J.A.K., and P.M.G. generated the main idea of the work and developed the study design. R.C.P., J.A.K., C.A.L., S.B.G., and P.M.G. performed the analysis and made interpretations of the data. Y.F., and G.B.W. made contributions to the acquisition of the data. J.B.H., J.M.G., and J.A.J. provided the LCLs for the project. M.M.W. maintained the LCLs. M.J.A., C.M., and P.M.G. supervised the data analysis. R.C.P., J.A.K., and P.M.G. wrote the manuscript from first draft to completion.

Y.F. and K.L.T. made comments, suggested appropriate modifications, and corrections. All authors read and approved the final manuscript.

## Additional information

**Competing interests:** The authors declare no competing interests.

