## [Peer Review File · Nature Communications]

Reviewer #1 (Remarks to the Author):

This manuscript presents novel data showing how genetic variants associated with autoimmune disease can influence histone epigenetic marks and gene expression, which will be of interest for researchers both within and outside the field.

The data and methodology are presented in an appropriate way.

Was the gene expression data presented obtained from the same individuals as the LCL used for the ChIP-Seq and HiChIP data? Some discussion on this would be needed.

Gene names should be written in italics

Page 8 line 174: please provide exact P value.

Reviewer #2 (Remarks to the Author):

In this manuscript, Pelikan et al. describe the identification of "epiQTLs" using a panel of 25 lymphoblastoid cell lines derived from individuals with SLE. They demonstrate that these "epiQTLs" are enriched in autoimmune disease risk loci and suggest these variants work in cooperation with eQTLs to explain additional variance in gene expression levels. Notably, they describe the enrichment of the epiQTLs in the HLA Class I and II loci and demonstrate that HLA haplotypes correspond changes in the 3-dimensional organization of the locus which may explain changes the observed differences in gene expression.

Overall, I found this study to be an interesting approach to delineating the genetic variants that contribute to gene dysregulation and clinical risk and appreciated the work to integrate eQTLs, epiQTLs, epigenetic data and GWAS results. Overall, I found some of the analyses to lack robustness. Additional tests, controls and replication, which I believe can reasonably be accomplished, would greatly strengthen this manuscript and the conclusions drawn herein. This study would also be greatly strengthened by further analysis of the HLA locus. The correlation of haplotype with reduced looping is an interesting finding, but how the variants within the haplotype contribute to this change in 3D chromatin dynamics is not discussed.

Major Points:

1) Replication: The rather large number of QTLs reported from a relatively small input data set raises the question of replication. Demonstrating the reproducibility of these results would greatly strengthen the manuscript. Previous studies have evaluated histone modification and DNase QTLs in LCLs. These data likely provide opportunity to assess reproducibility of the findings.

a. Kasowski M, et al. Extensive variation in chromatin states across humans. *Science*. 2013;342:750-2.

b. Kilpinen H, et al. Coordinated effects of sequence variation on DNA binding, chromatin structure, and transcription. *Science*. 2013;342:744-7.

c. Degner JF, et al. DNase I sensitivity QTLs are a major determinant of human expression variation. *Nature*. 2012;482:390-4

2) Effect sizes: The effects of the epiQTLs on allelic imbalance are not thoroughly described. It is unclear if the epiQTLs described are predominately small effects that shift allelic imbalance to 55/45 allelic ratios vs. 80/20 for example. Visualization of examples of epiQTLs are desired, i.e. dot/box plots of allelic ratios and/or browser images of described epiQTLs. What is the average effect size, what are the effect sizes for the strongest epiQTLs, or HLA epiQTLs compared to others. Etc.

a. Can the results described in Lines 107-115 be shown as well as told?

3) I would appreciate some clarification and details in the description of the epiQTL results.

a. The manuscript discusses the consensus peak list and the methods section describes analyzing 2-kb regions. Can you provide some additional clarity for how reported epiQTLs relate to distinct ChIP peaks. I.e. Are there multiple 2-kb regions per peak, are these reported as separate epiQTLs? Of the 6,261 epiQTLs reported, how many unique ChIP peaks have at least one significant epiQTL?

b. Are the 2,007 "independent" effects different enhancers/peak regions? Do you often have multiple SNPs not in LD impact the same putative enhancer?

c. Lines 115-117: Are the epiQTLs described as unique to the H3K27ac mark enriched for H3K4me1? If so, does this indicate you have greater power to detect promoter QTLs where H3K27ac levels are higher. Likewise of H3K4me1 specific epiQTLs are these regions enriched for H3K27ac? What percentage of epiQTL tests were in consensus peak lists for both marks? The description of epiQTLs being unique to one mark or the other is difficult to interpret without additional description.

4) DeepSEA analysis: I am not clear on what this analysis adds to this manuscript. These results could simply be descriptive of the putative enhancer elements active in LCLs. Are the identified

clusters unique to epiQTLs or would a random selection of from the consensus peak list yield the same results? The conclusion in line 130, "The distinguishing characteristics of these chromatin effect profiles highlight the functional influence of epiQTLs beyond the lymphoid LCL model" seems premature given the results demonstrated by DeepSEA.

5) Is the enrichment of risk loci for epiQTLs described lines 142-147 solely driven by the MHC locus? Are non-MHC epiQTLs enriched for autoimmune risk loci?

6) The results presented in Figure 2 are very interesting. I appreciate the effort to integrate epiQTL and eQTLs but have some concerns about the methods utilized. Which I believe can be addressed upon revision.

a. Are the changes in D2 observed significant? The permutation approach described appears to generate a null distribution of expected D2 values and compared this to the average increase in D2. A more stringent approach would be to generate a null distribution of D2 for each locus. Does the D2 for the lead epiQTL increase variance explained more than expected by including an additional variant (non- epiQTL) at random? This would enable determination of which loci the epiQTL significantly increases D2, and the number of loci for which including the epiQTL made a substantial impact of the ability to model gene expression. Given the number of loci that fall near the identity line, further analysis of the magnitude of the observed effects would help dissuade concerns that these impact is real.

b. A possibility not described in the manuscript is that the lead eQTL and lead epiQTL could be tagging a single causal eQTL (refs), rather than acting as two independent causes that impact target gene expression. This possibility is higher here given the high D' between the two variants. Likewise, if epiQTL is the actual causal variant underlying the eQTL signal that would also explain why including that variant in the model increased D2. Does modeling the two variants together explain significantly more variance in target gene expression than EITHER variant independently? The LD thresholding to evaluate variants with high D' in $r^2 < .8$ is not sufficient to demonstrate/conclude the variants are independent.

c. Do epiQTLs impact gene expression when considered independently? Could conditional analysis to be used to determine if the epiQTL and eQTL have independent contributions to gene expression?

7) HiChIP analysis of the HLA locus. This is a very interesting finding and the use of HiChIP enables high resolution analysis of these results. A couple of questions regarding these analyses.

a. The results in this section suggest that the haplotype leads to reduced looping in the HLA locus which in turn alters gene expression. How do the variant in this locus lead to this effect? The

epiQTLs described in figure 3 appear to be concentrated over HLA-DRB1. Do epiQTLs give any evidence to the mechanism by which the haplotype contributes to altered looping?

b. Lines 248-250: the suggestion that reduced CTCF loop strength leads to reduced H3K27ac loops is made. Are there variants in the haplotype that underlie the CTCF-CTCF proposed insulated neighborhood loop? Or are the variants underlying the H3K27ac peaks? What evidence supports the conclusion that the CTCF loop strength is the cause of the decrease in H3K27ac looping and not vice versa?

c. Can the results shown in supplemental figure 7 be quantified and/or added to figure 4. The aggregate analysis is a nice result but showing reproducibility of these results across 3 samples greatly improves confidence in the conclusion.

d. Are epiQTLs enriched in active promoter interactions in HiChIP datasets relative to non epiQTL enhancer loci?

8) A high number of significant interactions are reported. How many of these are redundant i.e. involve LD surrogates of one another? Previous studies have demonstrated that statistical artifacts often confound analysis of interacting variants. These concerns do not appear to be addressed in this analysis.

a. Are Interactions between cis-Regulatory Variants Evidence for Biological Epistasis or Statistical Artifacts? *Am J Hum Genet.* 2016 Oct 6;99(4):817-830. doi: 10.1016/j.ajhg.2016.07.022. Epub 2016 Sep 15.

b. Detection and replication of epistasis influencing transcription in humans. *Nature.* 2014 Apr 10;508(7495):249-53. doi: 10.1038/nature13005. Epub 2014 Feb 26.

c. Another explanation for apparent epistasis. *Nature.* 2014 Oct 2;514(7520):E3-5. doi: 10.1038/nature13691.

Minor points:

1) The term "peaks" is confusing in the context of epiQTLs. Distinguishing between ChIP/enhancer peaks and QTL signal "peaks" would aid overall clarity

2) Lines 174-176 Moreover knowing... 87% please clarify.

9) Figure 3, the demonstration of effect sizes and display of the H3K27ac ChIP track are greatly appreciated. A subpanel that zooms into the locus with 3 signals for DR15 and DR3 and displays LD between the variants would facilitate better understanding of the locus.

3) Figure text size is very small

4) Insulated neighborhoods and TADs lack citation

Reviewer #3 (Remarks to the Author):

The authors present a study on QTLs in LCLs of patients suffering from SLE in H3K27ac and H3K4me1 (which they term epiQTLs). Overall, they present that shows the importance of the HLA locus for auto immune disease risk alleles - which is not necessarily novel, what is novel though is the detailed molecular mechanism with specific chromatin loops being formed only in one of the two haplotypes. At times the manuscript is very difficult to read for non population-genetics people, which is a pity and should be addressed by the authors.

Major comments:

- in Figure 1C/D: it is not clear to me what the clusters are based on. Based on the underlying sequence? What are the e-values indicating? I think this should be made clear for readers that have not used DeepSEA before. Also a short description of what DeepSEA actually does and why it is an appropriate way to functionally characterise the epiQTLs.

- overlap with AI SNPs: the authors present a lot of numbers without referring to the number of SNPs that would be expected by chance to overlap with these loci

- it should also be mentioned what the background was that they used to calculate the enrichment since it is known that disease-associated SNPs fall into active regions the set of peaks should be taken as the background and SNPs should be matched for minor allele-frequency and proximity to transcription start sites.

- additionally it would be interesting to see whether this enrichment is indeed specific for auto immune diseases, or whether this is also seen for seemingly unrelated traits (in which case it is more likely that the active nature of the chromatin marks they look at is confounding the analysis)

- it is not clear why the reanalysis of the gEUVADIS eQTLs results in 26% less eQTLs than what the original study reported. The authors should comment on what is different between their reanalysis and the original eQTL calls and why they think their analysis is better.

- the part about the variants in strong LD ($D' \geq 0.8$) but not good proxies ($r^2 < 0.8$) is not understandable by the audience I assume the authors want to reach. These are very population

genetic-specific terms that should be explained in more detail in terms of what they mean. Also it is not clear whether the findings are specific to the particular thresholds that the authors chose for the two measures. I would at least expect some analyses with varying those thresholds to make sure the findings are robust to small changes in how LD and proxy was defined.

- it is also not clear in the same section how "non-epiQTLs" are chosen. Are they matched in terms of everything else (minor allele-frequency, distance to TSS, peak signal) to the set of epi-QTLs? Otherwise I think this comparison is invalid

In general, I'm not sure how much this particular section adds to the manuscript, the part that follows with HiC data and the HLA locus is much more straight-forward and might be missed by many readers that get frightened away by this very technical section that in my opinion doesn't add too much to the story. Might be worth changing the order.

I did not understand what the sentence: "The region telomeric to the shared peak..." on page 10. What is a region "telomeric to" what is a doublet epiQTL?

Minor comments:

- cluster in 1C, 1D: would be nice to get some examples of TFs that make the authors claim that this is the B-lymphoid cluster

- what the authors describe with epiQTLs has been termed histone (hQTLs) before. Unless there is a strong reason for changing the name I suggest to stick to the hQTL

- Fig4a: would be nice to indicate by a specific color which loops are specifically only found in DR3 or DR15

- Fig4c: would be good to indicate where the examples in 4C are located in either a) or b) (if they are at all)

Reviewer #1:

- 1) **Was the gene expression data presented obtained from the same individuals as the LCL used for the ChIP-Seq and HiChIP data? Some discussion on this would be needed.**

In order to maximize our power for the deviance and triplet analyses, we state in the manuscript that we utilized the gene expression data from the gEUVADIS Caucasian samples for these analyses. However, for the HLA gene expression data described in lines 245-251, we state that the RNA-seq data used to generate the gene expression data were collected on the same 25 LCL samples used for the ChIP-seq and HiChIP data. Data were used from the gEUVADIS Caucasian samples for replication (lines 251-254).

- 2) **Gene names should be written in italics.**

All gene names are now italicized in the manuscript.

- 3) **Page 8 line 174: please provide exact P value.**

P = 1.88E-8 is now included in the manuscript.

Reviewer #2:

- 1) **Replication: The rather large number of QTLs reported from a relatively small input data set raises the question of replication. Demonstrating the reproducibility of these results would greatly strengthen the manuscript. Previous studies have evaluated histone modification and DNase QTLs in LCLs. These data likely provide opportunity to assess reproducibility of the findings.**
 - a. Kasowski M, et al. Extensive variation in chromatin states across humans. *Science*. 2013;342:750-2.
 - b. Kilpinen H, et al. Coordinated effects of sequence variation on DNA binding, chromatin structure, and transcription. *Science*. 2013;342:744-7.
 - c. Degner JF, et al. DNase I sensitivity QTLs are a major determinant of human expression variation. *Nature*. 2012;482:390-4

We thank the reviewer for this suggestion and agree that reproducibility of our results has greatly strengthened our manuscript. Since our study sample is Caucasian (and to keep from introducing additional allele frequency variation from the Yorubans), we performed our replication study on the publicly available data from the combined, independent Caucasian 1000G phase 3 samples from the first two studies listed above (N=10). We have added the results to the manuscript, to Supplementary Table 1, and added a new Supplementary Table 2 and Supplementary Figure 2. In short, despite the replication sample being only 40% the size of our discovery sample and less than 1/3rd the sequencing read depth (a critical parameter for allele specific analyses), 4.6% (N=234) of the testable hQTLs produced $p < 1.0E-05$, with 43% overall (N=2,181) producing $p < 0.05$. We believe these findings exhibit strong support for reproducibility of our identified LCL hQTLs.

2) **Effect sizes: The effects of the epiQTLs on allelic imbalance are not thoroughly described. It is unclear if the epiQTLs described are predominately small effects that shift allelic imbalance to 55/45 allelic ratios vs. 80-20 for example.**

- **Visualization of examples of epiQTLs are desired, i.e. dot/box plots of allelic ratios and/or browser images of described epiQTLs. What is the average effect size, what are the effect sizes for the strongest epiQTLs, or HLA epiQTLs compared to others. Etc.**

We have now included a section in the paper describing the effect sizes of our hQTLs and have added a volcano plot to Figure 1, and the results to Supplementary Table 1 to better describe the effects of hQTLs on allelic imbalance. In short, the average hQTL $\log_2(\text{effect size})$ was 0.89. Strongest effects were observed with non-HLA variants (average $\log_2(\text{effect size}) = 0.90$) compared to HLA variants (average $\log_2(\text{effect size}) = 0.83$). These results demonstrate that our hQTLs produced strong effects on allelic imbalance with variants having, on average, almost two-fold more histone reads with one allele versus the other.

- **Can the results described in Lines 107-115 be shown as well as told?**

These results are presented as annotated Manhattan plots in Figures 1A & B and are referred to in the text. We have also added bold font in the annotation of the Manhattan plots to highlight those genes mentioned in the text.

3) **I would appreciate some clarification and details in the description of the epiQTL results:**

- **The manuscript discusses the consensus peak list and the methods section describes analyzing 2-kb regions. Can you provide some additional clarity for how reported epiQTLs relate to distinct ChIP peaks. ie:**
 - **Are there multiple 2-kb regions per peak, are these reported as separate epiQTLs?**

We apologize for the confusion. "Consensus peaks" or "peaks" are defined by the ChIP data while the 2-kb regions are scanned by the CHT software and are defined by the positions of the SNPs in the genotyping data; 1-kb scanned upstream and downstream of the SNP. If multiple SNPs are present within the same ChIP seq peak, then yes, there can be multiple SNPs within the same 2-kb region but they are reported as separate hQTLs.

- **Of the 6,261 epiQTLs reported, how many unique ChIP peaks have at least one significant epiQTL?**

Our hQTLs were confined to a small number of consensus peaks. The 4,858 hQTLs unique to the H3K27ac mark were located in only 879 (2.6%) H3K27ac consensus peaks while the 817 H3K7me1 hQTLs were confined to only 628 (1.6%) H3K27me1 consensus peaks. We have added this information to the manuscript.

- **Are the 2,007 "independent" effects different enhancers/peak regions? Do you often have multiple SNPs not in LD impact the same putative enhancer?**

We apologize for our previous error in describing the 2,007 as "independent" effects. We have 2,007 hQTLs that are not proxies ($r^2 < 0.8$) of another hQTL, but many of these 2,007 are still dependent of others based on D'. We have removed the word "independent" and have corrected the text to say the following: "In total, we identified 6,261 significant hQTLs (2,007 with $r^2 \leq 0.8$) distributed throughout the genome."

Yes, it is possible that we have multiple SNPs not in LD that impact the same putative enhancer. We have more than one SNP per enhancer ~36% of the time. While we describe in the paper the number of peaks that cover the total number of hQTL effects for each histone mark, when considering only the 2,007 effects with $r^2 < 0.08$, 1,386 hQTLs were unique to the H3K27ac mark and located in 522 of the H3K27ac consensus peaks, while 604 hQTLs were unique to H3K4me1 and found in 314 of the H3K4me1 consensus peaks.

- **Lines 115-117: Are the epiQTLs described as unique to the H3K27ac mark enriched for H3K4me1? If so, does this indicate you have greater power to detect promoter QTLs where H3K27ac levels are higher. Likewise of H3K4me1 specific epiQTLs are these regions enriched for H3K27ac?**

A total of 34% of H3K27ac specific hQTLs are enriched for H3K4me1 (also contain a H3K4me1 ChIP peak). A total of 27.4% of specific H3K4me1 specific hQTLs are enriched for H3K27ac. We do not believe these results indicate greater power to detect promoter QTLs where H3K27ac levels are higher because 1/3 of the effects still exhibited a H3K4me1 enhancer but the reads did not demonstrate allele specificity for the H3K4me1 mark at these loci. Our goal was not to specifically detect promoter QTLs, but rather to consider all enhancer peaks for these two marks to detect hQTLs.

- **What percentage of epiQTL tests were in consensus peak lists for both marks? The description of epiQTLs being unique to one mark or the other is difficult to interpret without additional description.**

Only 586 (9%) of hQTLs were in consensus peaks shared by both marks.

- 4) **DeepSEA analysis: I am not clear on what this analysis adds to this manuscript. These results could simply be descriptive of the putative enhancer elements active in LCLs. Are the identified clusters unique to epiQTLs or would a random selection of from the consensus peak list yield the same results? The conclusion in line 130, "The distinguishing characteristics of these chromatin effect profiles highlight the functional influence of epiQTLs beyond the lymphoid LCL model" seems premature given the results demonstrated by DeepSEA.**

We agree with the reviewer and have removed this figure and accompanying discussion from the text.

5) **Is the enrichment of risk loci for epiQTLs described lines 142-147 solely driven by the MHC locus? Are non-MHC epiQTLs enriched for autoimmune risk loci?**

We thank the reviewer for this question. We have now performed analyses with and without HLA and demonstrate that our results are not solely driven by the MHC locus. After performing permutation tests to determine if our observations were greater than what would be expected by chance, we found that our hQTLs were indeed enriched on haplotypes associated with AI disease (with HLA: $p_{\text{permutation}} = 4.9\text{E-}62$; $N = 386$ observed / 180 expected by chance; without HLA: $p_{\text{permutation}} = 4.4\text{E-}3$; $N = 177$ observed / 146 expected by chance), SLE (with HLA: $p_{\text{permutation}} = 1.9\text{E-}4$; $N = 68$ observed / 45 expected by chance; without HLA: $p_{\text{permutation}} = 0.073$; $n=48$ observed / 38 expected by chance), and complex traits reported in the NHGRI database (with HLA: $p_{\text{permutation}} = 2.1\text{E-}90$; $N = 1,520$ observed / 1007 expected by chance; without HLA: $p_{\text{permutation}} = 4.2\text{E-}40$; $n=1,151$ observed / 839 expected by chance). This information has been added to the text.

6) **The results presented in Figure 2 are very interesting. I appreciate the effort to integrate epiQTL and eQTLs but have some concerns about the methods utilized. Which I believe can be addressed upon revision.**

- **Are the changes in D2 observed significant? The permutation approach described appears to generate a null distribution of expected D2 values and compared this to the average increase in D2. A more stringent approach would be to generate a null distribution of D2 for each locus. Does the D2 for the lead epiQTL increase variance explained more than expected by including an additional variant (non- epiQTL) at random? This would enable determination of which loci the epiQTL significantly increases D2, and the number of loci for which including the epiQTL made a substantial impact of the ability to model gene expression. Given the number of loci that fall near the identity line, further analysis of the magnitude of the observed effects would help dissuade concerns that these impact is real.**

We are pleased that the reviewer found this section of the manuscript interesting and appreciate the suggestion for performing the analysis separately for each locus. As suggested, we repeated this analysis at a higher resolution by subjecting each eQTL locus to its own permutation-based analysis. On average, the change is significant, as initially reported. When examining individual loci, 36% of these loci now experience a significant increase in D^2 when incorporating genotypic information at the hQTL. The results are summarized in Figure 2.

- **A possibility not described in the manuscript is that the lead eQTL and lead epiQTL could be tagging a single causal eQTL, rather than acting as two independent causes that impact target gene expression. This possibility is higher here given the high D' between the two variants. Likewise, if epiQTL is the actual causal variant underlying the eQTL signal that would also explain why including that variant in the model increased D2. Does modeling the two variants together explain significantly more variance in target gene expression than EITHER variant independently? The LD**

thresholding to evaluate variants with high D' in $r^2 < 0.8$ is not sufficient to demonstrate/conclude the variants are independent.

We thank the reviewer for this thorough evaluation of our analyses and thought-provoking questions. We agree with the reviewer that $r^2 < 0.8$ is not sufficient to demonstrate that the variants are independent and have adjusted the analysis to require that the hQTL has $r^2 \leq 0.6$ with the lead eQTL. As the hQTL and lead eQTL are already in high LD, we don't wish to claim independence, but rather lack of genetic correlation between the two markers, so that both variants contribute different information within the context of strong LD. Secondly, in the loci analyzed here, we did not consider haplotypes where the lead hQTL was also the lead eQTL ($n = 76$). This makes it less likely that the hQTL will be the actual causal variant tagging the eQTL effect. Third, the gain in expression variance explained is almost never significantly more than modeling either variant separately. This is because eQTLs are selected *a priori* on their basis for being highly correlated with gene expression, whereas hQTLs are related to a different phenomenon – epigenetic activity – which is not always well-reflected by the eQTL target gene. In most cases the eQTL alone explains most of the expression variation, and the hQTL functions as a modifier of eQTL expression.

- **Do epiQTLs impact gene expression when considered independently? Could conditional analysis be used to determine if the epiQTL and eQTL have independent contributions to gene expression?**

Yes, we find that 522 (~8%) of our hQTLs impact gene expression when considered independently. However, the point of this analysis was not to examine the independent contribution of hQTLs. We wished to emphasize that hQTLs may be a modifier to causality in the context of strong LD, which, in a typical setting, will usually be identified by a disease risk variant that demonstrates an eQTL effect on a target gene(s).

7) HiChIP analysis of the HLA locus. This is a very interesting finding and the use of HiChIP enables high resolution analysis of these results. A couple of questions regarding these analyses.

- **The results in this section suggest that the haplotype leads to reduced looping in the HLA locus which in turn alters gene expression. How do the variant in this locus lead to this effect? The epiQTLs described in figure 3 appear to be concentrated over HLA-DRB1. Do epiQTLs give any evidence to the mechanism by which the haplotype contributes to altered looping?**

We thank the reviewer for his/her thorough evaluation of our HLA analyses. After working through the suggestions, we now recognize the necessity for aligning the DR3 and DR15 reads to their respective COX (HLA-DR3) and PGF (HLA-DR15) genomes and hope that this information is helpful for other researchers who wish to study this important locus. We found that aligning our DR3 subjects to the DR15 human reference genome biased our looping results, leading to reduced looping in the HLA class II locus for DR subjects. To address this, we completely revised this analysis using reference genome alignment to the specific HLA haplotype of our DR3 and DR15 subjects. With haplotype-specific alignment, we now have a more thorough and realistic picture of what is occurring within the

HLA and a comprehensive representation of chromatin interactions for both reference genomes. Importantly, our overall interpretations have not changed and the new analysis actually provides stronger evidence for the differential HLA-D gene expression we observe in the region. We found that subjects with the HLA-DR15 risk haplotype demonstrated significantly increased H3K27ac-mediated chromatin loop frequencies to enhancers flanking *HLA-DQA1* and *HLA-DQB1*. These enhancers correspond to regions of strong allelic imbalance and hQTLs observed in the HLA-DR15 subjects. Alternatively, enhancers located in the region producing the strongest hQTLs among the HLA-DR3 individuals produced unique loops to the promoter of *HLA-DRB1*. While the underlying mechanisms remain to be fully clarified, we believe the strong allelic imbalance at the hQTLs in this region is modifying the chromatin landscape and, thus, driving the differential gene expression between the two haplotypes.

- **Lines 248-250: the suggestion that reduced CTCF loop strength leads to reduced H3K27ac loops is made. Are there variants in the haplotype that underlie the CTCF-CTCF proposed insulated neighborhood loop? Or are the variants underlying the H3K27ac peaks? What evidence supports the conclusion that the CTCF loop strength is the cause of the decrease in H3K27ac looping and not vice versa?**

An additional benefit of our revised alignment of HiChIP data to haplotype-specific genomes indicates that CTCF loop strength is actually quite stable across both haplotypes. With the new analysis, we did not identify a direct effect of hQTLs on CTCF looping. We have removed the claim that reduced CTCF looping may govern H3K27ac looping.

- **Can the results shown in supplemental figure 7 be quantified and/or added to figure 4. The aggregate analysis is a nice result but showing reproducibility of these results across 3 samples greatly improves confidence in the conclusion.**

Thank you for this suggestion. Figure 4 now includes panel 4C, which shows the reproducibility of the looping across the 6 individuals demonstrating the looping differences in each subject.

- **Are epiQTLs enriched in active promoter interactions in HiChIP datasets relative to non epiQTL enhancer loci?**

A total of 87% of our hQTLs are involved in enhancer-promoter looping events.

- 8) **A high number of significant interactions are reported. How many of these are redundant i.e. involve LD surrogates of one another? Previous studies have demonstrated that statistical artifacts often confound analysis of interacting variants. These concerns do not appear to be addressed in this analysis.**
- a. Are Interactions between cis-Regulatory Variants Evidence for Biological Epistasis or Statistical Artifacts? Am J Hum Genet. 2016 Oct 6;99(4):817-830. doi: 10.1016/j.ajhg.2016.07.022. Epub 2016 Sep 15.**
 - b. Detection and replication of epistasis influencing transcription in humans. Nature. 2014 Apr 10;508(7495):249-53. doi: 10.1038/nature13005. Epub 2014 Feb**

26.

c. Another explanation for apparent epistasis. *Nature*. 2014 Oct 2;514(7520):E3-5. doi: 10.1038/nature13691.

The reviewer is correct that our previous analysis did include some LD surrogates. While we removed all hQTL proxies ($r^2 > 0.6$) and required the eQTL and hQTL to be located on separate haplotypes, we did not initially prune the eQTL list and remove all proxies ($r^2 > 0.6$). We have since removed 8,372 interactions that involved 1,702 eQTL proxies ($r^2 > 0.6$) and have chosen to keep the hQTL demonstrating the greatest significant interactions when proxies were present. All of the numbers in the text have been updated.

Minor points:

- 1) **The term "peaks" is confusing in the context of epiQTLs. Distinguishing between ChIP/enhancer peaks and QTL signal "peaks" would aid overall clarity**

We apologize for this confusion. We have removed all reference to "peaks" when discussing the hQTL signals and hope this has resolved the issue.

- 2) **Lines 174-176 Moveover knowing... 87% please clarify.**

We have removed this text from the manuscript.

- 3) **Figure 3, the demonstration of effect sizes and display of the H3K27ac ChIP track are greatly appreciated. A subpanel that zooms into the locus with 3 signals for DR15 and DR3 and displays LD between the variants would facilitate better understanding of the locus.**

We thank the reviewer for this suggestion and agree that it provides a better understanding of the locus. We have now included a subpanel (Figure 3B) that zooms into this locus and have added a new Supplementary Figure 5 that provides the information for the LD in the region.

- 4) **Figure text size is very small**

Yes, we recognize this issue and have enlarged the text where feasible.

- 5) **Insulated neighborhoods and TADs lack citation**

The manuscript now includes citations for insulated neighborhoods and TADs.

Reviewer #3:

At times the manuscript is very difficult to read for non population-genetics people, which is a pity and should be addressed by the authors.

Major comments:

1) **In Figure 1C/D:**

- **it is not clear to me what the clusters are based on. Based on the underlying sequence?**

We have chosen to remove the DeepSea results and discussion from the manuscript.

- **What are the e-values indicating? I think this should be made clear for readers that have not used DeepSEA before.**

No longer applicable.

- **Also a short description of what DeepSEA actually does and why it is an appropriate way to functionally characterize the epiQTLs.**

No longer applicable.

2) **Overlap with AI SNPs:**

- **the authors present a lot of numbers without referring to the number of SNPs that would be expected by chance to overlap with these loci**

We have now conducted 1,000 permutation tests to determine if our observed enrichments were more than what would be expected by chance and discovered that our hQTLs are indeed enriched on haplotypes associated AI disease ($p_{\text{permutation}} = 4.9\text{E-}62$; $n=180$ expected by chance), SLE ($p_{\text{permutation}} = 1.9\text{E-}4$; $n=45$ expected by chance), and complex traits reported in the NHGRI database ($p_{\text{permutation}} = 2.1\text{E-}90$; $n=1,007$ expected by chance). We have added this information to the manuscript.

- **it should also be mentioned what the background was that they used to calculate the enrichment since it is known that disease-associated SNPs fall into active regions the set of peaks should be taken as the background and SNPs should be matched for minor allele-frequency and proximity to transcription start sites.**

We recognize and apologize that we did not initially include these methods in the submitted manuscript, which is now rectified. We have now taken into account the set of peaks as background and the SNPs have been matched for MAF and proximity to TSS.

- **additionally it would be interesting to see whether this enrichment is indeed specific for auto immune diseases, or whether this is also seen for seemingly unrelated traits (in which case it is more likely that the active nature of the chromatin marks they look at is confounding the analysis)**

We agree with the reviewer; however, the focus of our work is on autoimmune disease and since we demonstrate significant enrichment in these loci, we believe that this is sufficient to demonstrate the points we want to make.

- 3) **It is not clear why the reanalysis of the gEUVADIS eQTLs results in 26% less eQTLs than what the original study reported. The authors should comment on what is different between their reanalysis and the original eQTL calls and why they think their analysis is better.**

We have now added detail to the methods to describe the differences in the two analyses. Basically, the differences include the following: 1) while we attempted to use as many of the same analysis tools used by gEUVADIS, RNA-sequencing alignment tools have rapidly evolved and improved since the original gEUVADIS paper that utilized GEM. We, therefore, decided to use Kallisto and tximport for the alignment of RNA-sequencing reads. 2) The original paper included 373 samples from Phase 1 of the 1000 Genomes Project in the analysis; due to sample dropout in the Phase 3 genotyping data, publicly available data included only 358 samples.

- 4) **The part about the variants in strong LD ($D' \geq 0.8$) but not good proxies ($r^2 < 0.8$) is not understandable by the audience I assume the authors want to reach. These are very population genetic-specific terms that should be explained in more detail in terms of what they mean.**

We have changed the wording, which now reads, “Using these eQTLs as functional surrogates of risk haplotypes, we identified 268 haplotypes that contained both a gEUVADIS eQTL and one of our hQTL variants that were in strong LD ($D' \geq 0.8$) with each other but not highly correlated ($r^2 < 0.6$) in the genetic information they captured”.

- **Also it is not clear whether the findings are specific to the particular thresholds that the authors chose for the two measures. I would at least expect some analyses with varying those thresholds to make sure the findings are robust to small changes in how LD and proxy was defined.**

Yes, the findings are specific to the particular thresholds that we chose for D' and r^2 – that is the case with data analysis when thresholds are set. However, the values we used are common thresholds used by other studies in the literature and, therefore, we believe they represent acceptable parameters for this study.

- **It is also not clear in the same section how "non-epiQTLs" are chosen. Are they matched in terms of everything else (minor allele-frequency, distance to TSS, peak signal) to the set of epi-QTLs? Otherwise I think this comparison is invalid**

The analysis was performed separately for each haplotype block (defined by $D' \geq 0.8$) that contained an eQTL and an hQTL. Therefore, the “non-hQTL” variants were all of the non-eQTL and non-hQTL variants carried by the individual haplotype block that was being studied. Because all variants within the block were defined by strong levels of LD, the minor allele-frequencies, distance to TSS, etc were very similar to the hQTL they were tested with.

- **In general, I'm not sure how much this particular section adds to the manuscript, the part that follows with HiC data and the HLA locus is much more straight-forward and might be missed by many readers that get**

frightened away by this very technical section that in my opinion doesn't add too much to the story. Might be worth changing the order.

We appreciate the comments of the reviewer and have tried to improve the readability for non-geneticist readers. However, we feel the current arrangement of the sections fits best with our intended presentation of the data.

- 5) I did not understand what the sentence: "The region telomeric to the shared peak..." on page 10. What is a region "telomeric to" what is a doublet epiQTL?**

Since the HLA region is on the short arm of chromosome 6, "telomeric" refers to "towards the telomere" or, in this case, upstream of the shared signal. "Doublet" refers to the hQTL signal that has two peaks in close proximity to each other. We understand that this language could cause some confusion, so we have removed "telomeric" and have added the bp positions for the different signals for clarification.

Minor comments:

- 1) cluster in 1C, 1D: would be nice to get some examples of TFs that make the authors claim that this is the B-lymphoid cluster**

We have removed this figure and accompanying discussion in the text.

- 2) what the authors describe with epiQTLs has been termed histone (hQTLs) before. Unless there is a strong reason for changing the name I suggest to stick to the hQTL**

We now use the suggested hQTL terminology throughout the revised manuscript.

- 3) Fig4a: would be nice to indicate by a specific color which loops are specifically only found in DR3 or DR15**

We have now colored the loops that are unique to a particular haplotype.

- 4) Fig4c: would be good to indicate where the examples in 4C are located in either a) or b) (if they are at all)**

We have removed our previously submitted Figure 4C with the new haplotype specific alignment.

Reviewer #1 (Remarks to the Author):

The authors have responded to my comments in a satisfactory way.

Reviewer #2 (Remarks to the Author):

Overall, the authors have responded to my comments with detailed and largely sufficient changes or responses. In particular the addition of replication analysis and re-analysis of the HLA locus greatly strengthens the manuscript. Overall, I would suggest some strategic streamlining to the manuscript and to aid readability. I appreciate the approach described in lines 151-164, but find the section to lack clarity/readability.

Reviewer #3 (Remarks to the Author):

The authors have addressed most of the comments I had and provide clarifications and good arguments for those that were not addressed.